# Rainfall-runoff relationships at event scale in western Mediterranean ephemeral streams

Roberto Serrano-Notivoli[1], Alberto Martínez-Salvador[2], Rafael García Lorenzo[2], David Espín-Sánchez[2], Carmelo Conesa-García[2]

[1]Departamento de Geografía, Universidad Autónoma de Madrid, Madrid, 28049, Spain
[2]Departamento de Geografía, Universidad de Murcia, CEIR Campus Mare Nostrum (CMN), Campus de La Merced, Murcia, 30001, Spain

*Correspondence to*: Roberto Serrano-Notivoli (roberto.serrano@uam.es)

**Abstract.**Ephemeral streams are highly dependent on rainfall and terrain characteristics and, therefore, very sensitive to minor changes in these environments. Western Mediterranean area exhibits a highly irregular precipitation regime with a great variety of rainfall events driving the flow generation on intermittent watercourses, and future climate change scenarios depict a lower magnitude and higher intensity of precipitation in this area, potentially leading to severe changes in flows. We explored the rainfall-runoff relationships in two semiarid watersheds in southern Spain (Algeciras and Upper Mula) to model the different types of rainfall events required to generate new flow in both intermittent streams. We used a nonlinear approach through Generalized Additive Models at event scale in terms of magnitude, duration, and intensity, contextualizing resulting thresholds in a long-term perspective through the calculation of return periods. Results showed that the average ~1.2-day and <1.5 mm event was not enough to create new flows. At least a 4-day event ranging from 4 to 20 mm, depending on the watershed was needed to ensure new flow at a high probability (95%). While these thresholds represented low return periods, the great irregularity of annual precipitation and rainfall characteristics, makes prediction highly uncertain. Almost a third part of the rainfall events resulted in similar or lower flow than previous day, emphasizing the importance oflithological and terrain characteristics that lead to differences in flow generation between the watersheds.

## 1 Introduction

Precipitation plays a paramount role on drainage of the watersheds, especially in those depending on rainfall for the persistence of the flows, consideredintermittent streams. These types of watercourses, occasionally dry, are already a large-scale phenomenon (Acuña et al., 2005; Larned et al., 2010; Datry et al., 2014) and could be potentially increased under climate change conditions (Nabih et al., 2021; Brunner et al., 2020; Skoulikidis et al., 2017; Brooks, 2009). Thus, intensity and magnitude of rainfall events are a key part of hydrological models for simulation and prediction of floods in these

watersheds (Gioia et al., 2008; Kirkby et al., 2005) and knowing the thresholds required to generate new flows helps to tackle with natural hazards from a hydrological modelling perspective (Kampf et al., 2018).

Ephemeral streams are drainage networks remaining completely dry during a variable period of the year and, owing to rainfall events of certain magnitude, they can discharge relatively high flows (Donglioni et al., 2015) that can persist for some time. Western Mediterranean area is especially prone to accommodate watersheds with these types of streams because

of the high irregularity of precipitation, both in space and time (Tockner et al., 2009; Datry et al., 2017). In ephemeral streams, this irregularity turns into a great uncertainty in flow generation affecting not only the stream but also to other parts of the system. For example, the fickleness of flows alters the actual ecological functioning of the watershed at variable scales and, of course, affects the agricultural systems covering lowlands, that usually require infrastructures to retain water. Understanding how these watersheds react to precipitation is fundamental for prediction and forecasting of droughts and

floods (Döll and Schmied, 2012; Arnone et al., 2020), but also for erosion potentiality depending on the type of lithology under the soil and the type of vegetation or land cover at surface, and for sediment transport assessment (Fortesa et al., 2021). Previous research in ephemeral watersheds on Western Mediterranean (e.g., Camarasa and Tilford, 2002; Camarasa, 2016) showed that rainfall-runoff relationships drive hydrological processes and the dynamics of the rest of the system at basin scale, and that they can be modelled to forecast flows based on the rainfall events of different magnitude. These studies

highlight that, in the current Spanish Mediterranean scenario of decrease of total amounts of precipitation but increase in intensity (Serrano-Notivoli et al., 2018), hydrological connectivity is more dependent on rain intensity than in the past.

In this work, we explore the rainfall-runoff relationships in two watersheds with ephemeral streams in southeastern Spain: Algeciras (44.9 km$^2$) and Mula (169.4 km$^2$). Daily precipitation and flows from 17 and 24 years, respectively, were analysed at event scale to model the influence of rainfall events in the generation of new runoff in both watersheds. Due to the great

irregularity of precipitation, we used a nonlinear approach through Generalized Additive Models, and we compared the results in a wider temporal perspective through the calculation of return levels for several return periods. Based on the watershed physical and climatic characteristics, we hypothesise that runoff highly dependson the intensity and amount of rainfall of singular events.

## 2 Study site

The watersheds of Algeciras and Upper Mula are located within the semiarid climate characterizing the southeastern area of the Iberian Peninsula (Figure 1). Annual precipitation, with a manifest equinoctial regime (maximums in March-April and September-October) rarely exceeds 300 mm (Serrano-Notivoli et al., 2017), depicting the driest place in continental Europe. Average temperatures range from 10 to 26 ºC, however, temperatures above 30 ºC are common during summertime and absolute values higher than 40 ºC are not an exception (Serrano-Notivoli et al., 2019). With more than 100 days above 25 ºC,

the evapotranspiration rate is among the highest in Spain (Tomás-Burguera et al., 2020), leading to a negative water balance in the whole region, especially in summer months (June, July, and August) and being highly variable depending on the

season and the year. This water balance is sometimes aggravated by types of soil with high rates of infiltration, hampering surface runoff during most of the year

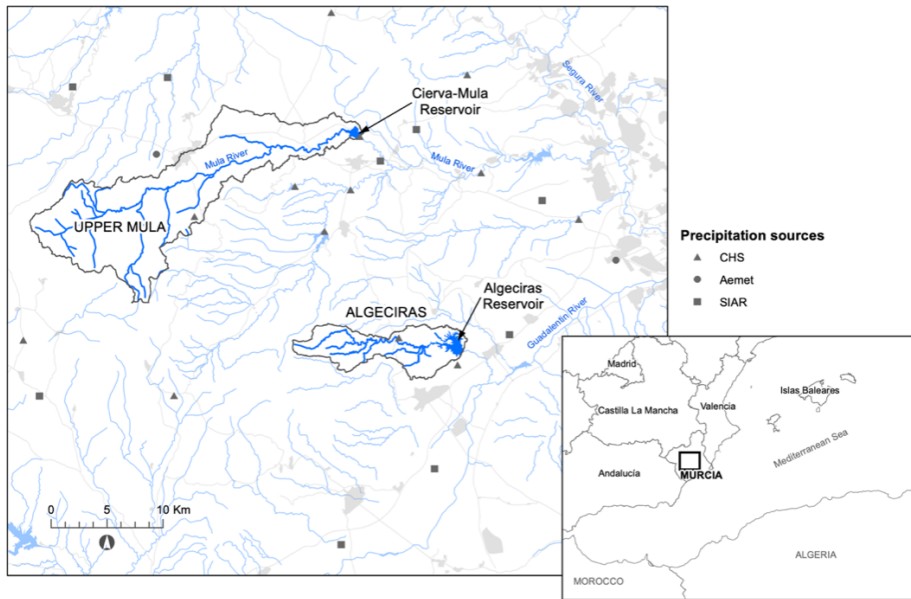

**Figure 1: Location of the watersheds and precipitation gauges**

The Upper Mula stream is an intermittenttributary at headwaters of the Mula River, which directly flows into the Segura River. Algeciras stream is an ephemeral watercourse draining into the Guadalentín River, the main tributary of the Segura River. Both basins belong to the geomorphological Betic and Subbeticdomain. Limestone and dolomites, sandstones,
siliceous marls, and detrital limestones predominate in their headwaters. However, their middle and lower parts are lithologically quite contrasted: marls and alluvial sediments are abundant in the Algeciras watershed, promoting a badlands landscape, while sandstone, conglomerates and detrital limestones predominate in the Upper Mula basin(Figure 2a and 2b). The land cover in the Algeciras is mainly composed of forest (28%), bare soil (25%) and scrubland (24%), while forest (39%), agricultural row crop (25%) and shrubland (20%) are dominant in the Upper Mula catchment (Figure 2c and 2d).
Lowlands of the watersheds are occupied by two reservoirs: Cierva-Mula (1929) and Algeciras (1995), both with a defensive function against floods and for irrigation control.

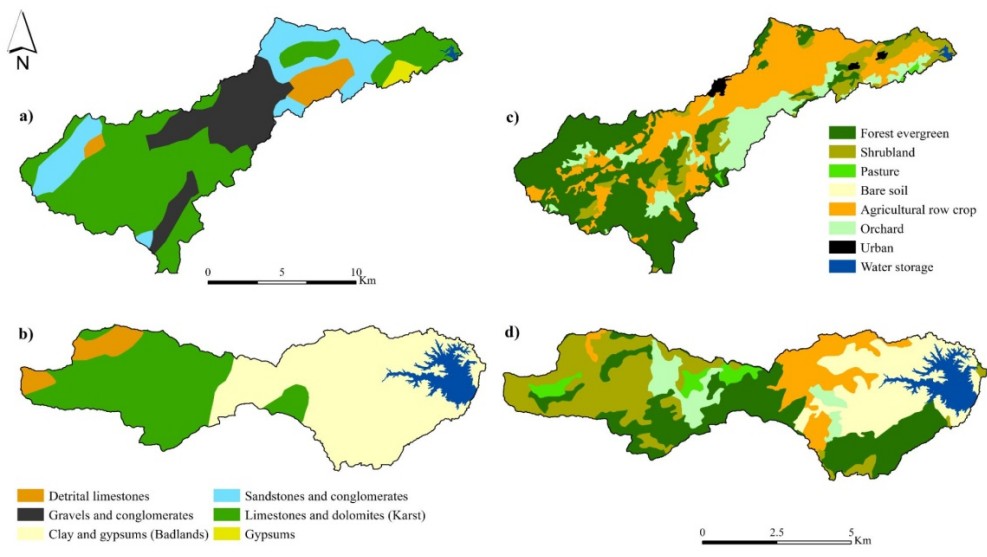

**Figure 2: Rock typesand land use in the Upper Mula (a,c) and Algeciras watersheds (b, d).**

## 2 Data and methods

### 2.1 Data

The data series of flows were obtained from the gauging reports supplied by the Center for Public Works Studies and Experimentation (CEDEX) for the Segura basin. We used the data series of the daily average flow ($m^3$/s) corresponding to periods 2003–2020 (Algeciras) and 1996–2020 (Upper Mula). Although Algeciras and Mula watersheds are ungauged and there are not direct measures of water discharge, the daily flow series were calculated from the difference between the volume of water stored in the reservoirs and the output of the previous day (Eq. 1).

$$E = (R - R_1) + S \qquad (1)$$

where $E$ is the inflow into the reservoir ($m^3$); $R$, the reserve of the current day ($m^3$); $R_1$, the reserve of the previous day ($m^3$); and $S$, the output flow of the previous day ($m^3$). While resulting daily series are not a direct measure of the streamflow, they provide the only representation of daily flow variations.

In order to provide single daily precipitation ($P$) series for each watershed, we created two regional series based on the information of meteorological stations (13 for Algeciras and 14 for Mula) from the Spanish meteorological agency (Aemet), the Agroclimatic Information System (SIAR) of the Spanish Ministry of Agrifood and Fisheries, and the Segura Hydrographic Confederation (CHS) (Figure 1). The regional series for each watershed were built with 2 variables: 1) the daily average of total precipitation in 24hours and 2) the daily average of maximum precipitation in 1 hour. With the aim of relating these series with the temporal availability of flow data, they were built for 2003-2020 in Algeciras and for 1996-

2020 in Mula. The original data series of the meteorological stations provided a representation of the real magnitude of precipitation events. Although the use of a spatial interpolation scheme had been useful to look for precipitation differences in a different situation (e.g., larger spatial domain, longer temporal period), the small extent of the study area (approx. 50x50 km) and the watersheds, along with the sizeable number of available observations, made the mean daily precipitation an average representation of the precipitation regime at event scale. In addition, the availability of single flow data series for each watershed constrained the analysis to a comparison with unique precipitation series. The complete process resulted in 2 series of daily precipitation and 2 series of hourly maximums in the same period of flows data series. Due to the reduced study area, most of the stations have a similar behaviour regarding precipitation occurrence, however, we considered as dry days those averaging a value lower than the minimum registered by the precipitation gauges (0.1 mm). The series of hourly maximums were built by averaging, for each day in all stations, the maximum precipitation cumulated in one hour. Despite the potential difference between stations, this measure represents the average intensity of daily precipitation. Lastly, we used the SPREAD dataset (Serrano-Notivoli et al., 2017), a daily gridded precipitation dataset covering the whole Spanish territory at a 5x5 km spatial resolution, to analyse long-term trends of annual precipitation of the two watersheds by extending its period coverage until 2020 following Serrano-Notivoli et al. (2017b). This analysis helped to study the low-frequency climatic signal of a broader spatial domain, by contextualizing the study period of each watershed since mid-20$^{th}$ century.

## 2.2 Statistical analyses at event scale

Instead of relating daily precipitation (*P*) with daily flows (*Q*), we opted to work at event scale due to consecutive wet days (*P > 0*) can have a different and more persistent impact on flow generation than single wet days. Rainfall events (*RE*) were detected from daily data series for the whole period in both watersheds by grouping consecutive wet days separated, at least, by one dry day (*P = 0*). We then calculated 4 variables for each event: duration (number of days); magnitude (sum of precipitation of all days); maximum (sum of hourly maximums of all days, to be representative of the amount of precipitation corresponding to the hours of maximum rainfall); and flow contribution (*ΔQ*, difference between the cumulated flow during the *RE* and flow of the day before the *RE*).

These variables were used to model the required characteristics of a *RE* to generate new flow at different probabilities on both watersheds based on:

1) the modeling of the rainfall-runoff response to identifywhich variables (duration, magnitude, or hourly maximums) and to what extent they contributed to flow generation at different probabilities; and

2) the calculation of the return periods of these contributing variables to estimate the likelihood of occurrence of(highest probabilities) of flow generation.

### 2.2.1 Rainfall-runoff modelling

We performed, using all events, a simple linear correlation analysis between the four variables for an overview of the general linkage between each other. However, ephemeral streams involve highly nonlinear relationships between rainfall and runoff (Ye et al., 1997) and, for this reason, we used Generalized Additive Models (GAMs) to detect further responses of the flows to rainfall at event scale. GAMs allowed for assessing simultaneous smooth relationships that can be linear or nonlinear as demonstrated in previous research (e.g., van Ogtrop et al., 2011). As the objective was to find out what type of event was necessary to generate flow in both basins, we used as dependent variable the *ΔQ* codified as a binomial variable (*Qbin*, *ΔQ>0*: *1*; *ΔQ<=0*: *0*) and duration, magnitude and maximum were treated as smooth predictor variables, specified using shrinkage smoothers (thin plate regression spline). GAMs were used with the logit link and the three variables were included in the model to predict *Qbin*, first individually, and then in combination with each other. All the models were compared, and the basis dimension of each smooth term was checked and increased when necessary. With the aim of evaluating the model accuracy with the selection of the best combination of variables for each watershed, we compared different models using from one to all variables through two conventional estimate errors (see Table A1): AIC (Akaike Information Criterion) and logLik (log-likelihood), and two specific estimate errors for GAMs: deviance (Residual deviance) and UBRE (Un-Biased Risk Estimator).Residual deviance is defined as twice the difference between the log likelihood of a model that provides a perfect fit (also called the saturated model) for the model under study (Zuur et al., 2009), andthe UBRE is essentially a rescaled AIC used to estimate the mean square error on GAMs (Wood, 2017).Concurvity (the analogue of multi-collinearity in GAMs) was tested in the final model (Table A2). To evaluate the hit rate of the models, we used a random sample of 75% of the RE in each watershed to set up the models. Then, predictions were computed for the remaining 25% and classified as probabilities from 0 to 1 as *P<0.5: 0* and *P>=0.5: 1* to be compared with the observations. A contingency table summarizing the hit rate helped to assess the model performance.

### 2.2.2 Return periods of highest probabilities of flow generation

To contextualize the *RE*required for different probabilities of generating flow in both watersheds, we estimated the return levels of their magnitude and maximums using apeak-over-threshold (POT) approach. POT is most suitable when complete time series (as *RE*) are available due to all values exceeding a certain threshold can serve as basis for model fitting (Coles, 2001).The objective was to estimate the return levels of magnitude and maximums of RE for different return periods. The POT method consists in fitting the *RE* observations higher than a specific threshold to a Generalized Pareto Distribution (GPD). The selection of this threshold must help to subset the appropriate number of observations to reduce the variance without choosing a too low threshold that could induce bias (Ribatet, 2007). In this case, the threshold was derived from the graphical representation of four parameters derived from the *RE* data: 1) the Mean Residual Life, which shows the mean value of observations over a threshold (mean excess). It is expected to be linear over the threshold at which GPD becomes valid (Acero et al., 2018); 2) the Dispersion Index, which is the ratio between variance and mean of the values over a

threshold, with an ideal theoretical value of 1; and 3) the modified scale and 4) shape parameters against a range of thresholds. The parameter estimates (3 and 4) are stable above the threshold at which the GPD model becomes valid. While interpretation of the plots is not always easy, we selected the appropriate thresholds (Figure A1 and A2) based on their convergence to the optimal values of the four graphical representations, as done in similar situations in previous works (Anagnostopoulou and Tolika, 2012; Zakaria et al., 2017).

Once thresholds were defined, We used four different estimators to fit the POT data to a GPD (Maximum Likelihood Estimation (MLE); Unbiased Probability Weighted Moments (PWMU); Moments (MOM); and Likelihood Moment (LME)) to establish proper and wide confidence levels in the estimate of maximum rainfall per *RE*.

## 3 Results

### 3.1 Characteristics of flows and precipitation

Average daily flows ($Q$) in Algeciras and Mula were relatively low in both watersheds (0.29 and 0.15 m$^3$/s, respectively) and these values were distant from the median of each month (Figure 3), denoting their great irregularity. However, the specific flow, that considers the size of the watershed, is 6.5 l/s/km$^2$ in Algeciras and 0.9 l/s/km$^2$ in Upper Mula (Table 1). Both watersheds had a similar precipitation regime, with a clear minimum in summer, especially in July, and maximums in spring and autumn (March and September are the rainiest months, respectively). However, their flows did not respond in the same

way to precipitation. While Mula had a more direct response to incident rainfall, Algeciras showed a different behaviour with their maximums at the end of summer and the beginning of autumn, associated to very high precipitation events. Also, the middle and lower parts of the Algeciras watershed are mainly covered with marls and alluvial sediments, creating an arid landscape consisting of a predominance of badlands and bare soil, where the rates of saturated hydraulic conductivity and hydraulic conductivity of the main channel are very low. Additionally, Algeciras show a higher curve number and slope than

Upper Mula and shorter concentration and lag times (Table 1).Thus, terrain characteristics play a key role on rainfall-runoff relationships, but also to the amount of Q per month. For instance, Mula have an average 30% more days per month with Q>0 than Algeciras, reaching almost 50% in summertime.

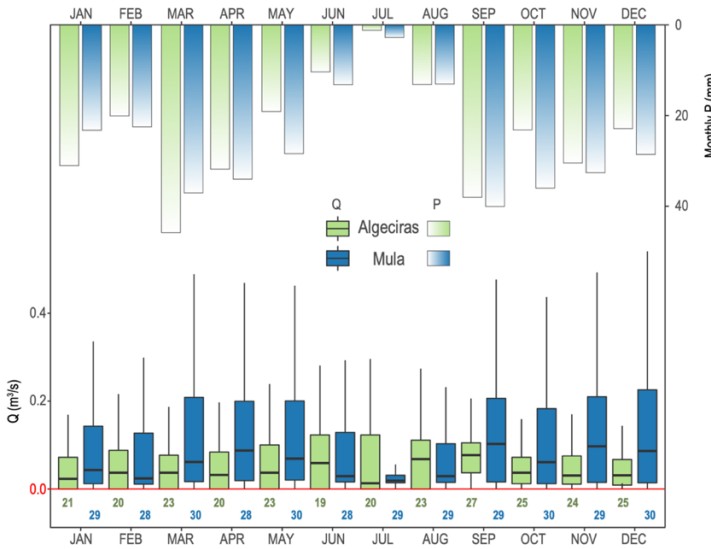

**Figure 3: Frequency of daily flows ($Q$) by month indicating low and high quantiles. Boxes show 25th to 75th percentiles with median as a bold horizontal line. Vertical lines reach 95th percentile (outliers are not shown). Bottom numbers show the mean number of days with $Q>0$. Bars from top indicate mean monthly precipitation ($P$).**

**Table 1: Geometric data of Algeciras and Mula watersheds.**

|  | Area | Longest stream | Stream slope | Watershed slope | Curve Number | Concentration time–Kirpich | Lag time | Specific flow |
|---|---|---|---|---|---|---|---|---|
| Algeciras | 44.9 km$^2$ | 25.1 km | 4.2% | 35.6% | 86.4 | 3.75 h | 2.25 h | 6.5 l/s/km$^2$ |
| Mula | 169.4km$^2$ | 45.5 km | 1.7% | 22.2% | 81.6 | 9.38 h | 5.63 h | 0.9 l/s/km$^2$ |

### 3.1.1 Rainfall events (RE) over time

The long-term analysis of annual precipitation showed different behaviours of the watersheds in the first two decades of 21$^{st}$ century (Figure 4) than in previous periods, coinciding with the period of study (when available flow data series). Algeciras showed a higher frequency of drier years until the end of 1980s' decade. Then, this pattern changed and 13 of the first 20 years of 21$^{st}$ century were wetter than the average, concurring a positive anomaly of the number of precipitation days. Linear trend indicated a non-significant increase of 7.2 mm/decade of annual precipitation and a significant increase of 7.1 days/decade of number of wet days per year. In summary, Algeciras experienced an increase of precipitation events with an uncertain increase of their magnitude. However, precipitation amounts in 2000-2020 period were significantly lower than the three previous decades.

The irregularity of annual precipitation in Mula provided an also irregular depiction of its anomalies through time. While 1950-1970 period showed a rotation of wet and dry years, the decade of 1970 was the wettest and, since then, most of the years were below the average precipitation. The anomaly of wet days showed a regular behaviour from 1960 to 2000, when they increased until 2020. Precipitation amounts showed a negative and non-significant trend of 8.6 mm/decade and a positive significant trend of number of wet days of 7.8 days/decade.

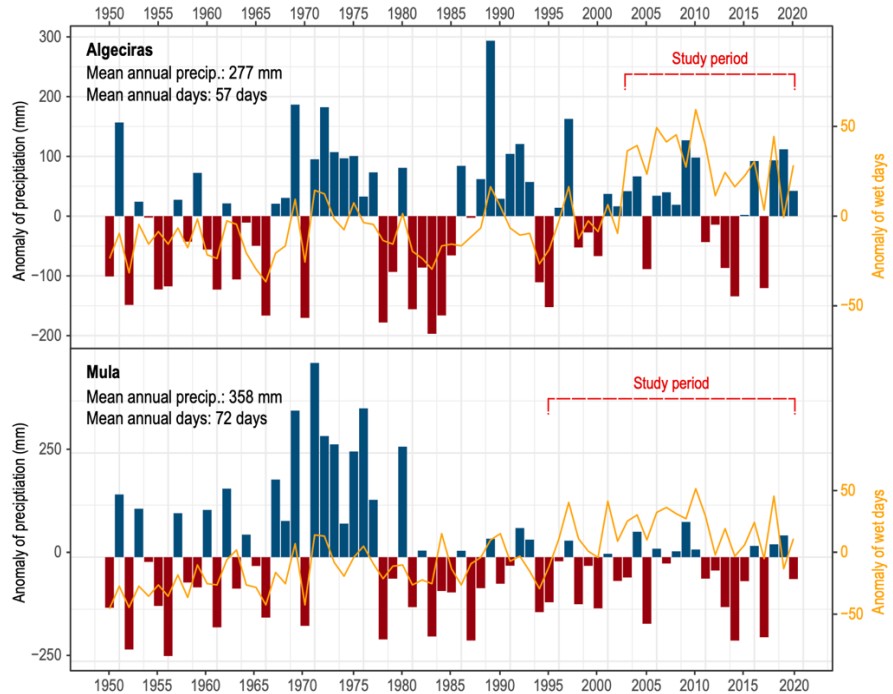

**Figure 4: Annual precipitation anomalies (bars) and annual anomaly of the number of wet days (*P>0*) (lines). 1950-2020 was used as base period. Dashed lines indicate the period of data used for the analysis, coinciding with flows data availability.**

When analysing the study periods at event scale (Figure 5), both watersheds showed most of the highest magnitudes of precipitation in 2019 and 2020. While Algeciras showed a more regular response of flow contribution (*ΔQ*) to RE throughout the study period, Mula experienced high *ΔQ* in high magnitude events until 2000. Then, the response was faster, with similar (or higher) magnitude events and lower *ΔQ* than in the previous period. The duration of RE was varied in both watersheds, and not always long event resulted in a high magnitude of precipitation and a high *ΔQ*. In fact, the frequency of high-

magnitude events was higher from 2016 in Algeciras and Mula, but it was not accompanied by longer durations.

A non-negligible proportion of RE produced a zero (14% in Algeciras and 3% in Mula) or negative (22% and 23%) *ΔQ*, meaning that the flow contributed resulted in a similar or lower value than the previous day of the event, respectively. These RE, that were very similar in both watersheds, were short and small in terms of amount of rainfall. With a mean magnitude between 0.5 and 1.5 mm and a mean duration from 1.2 to 1.3 days, the generation of new flow is difficult. The reason of why

these RE did not produced any flow contribution are related to the flow and precipitation regimes of the watersheds. For instance, a large proportion of non-contributing RE were from June to August (Table 1), the months with lowest precipitation, the lowest number of days with *Q>0* (Figure 2), and the highest evapotranspiration (Tomás-Burguera et al., 2020). Algeciras showed 10 months with proportions higher than 30%, a large difference compared to Mula (4 months), and this is also explained by the higher intermittency of Algeciras stream. Also, the geomorphological characteristics of the

watersheds play a fundamental role on the *ΔQ*: small RE in combination with unsealed and fragile soils favour the

infiltration (limestone lithologies prevail in Mula) and, especially in summer, evaporation, which necessarily leads to the absence of new flows.

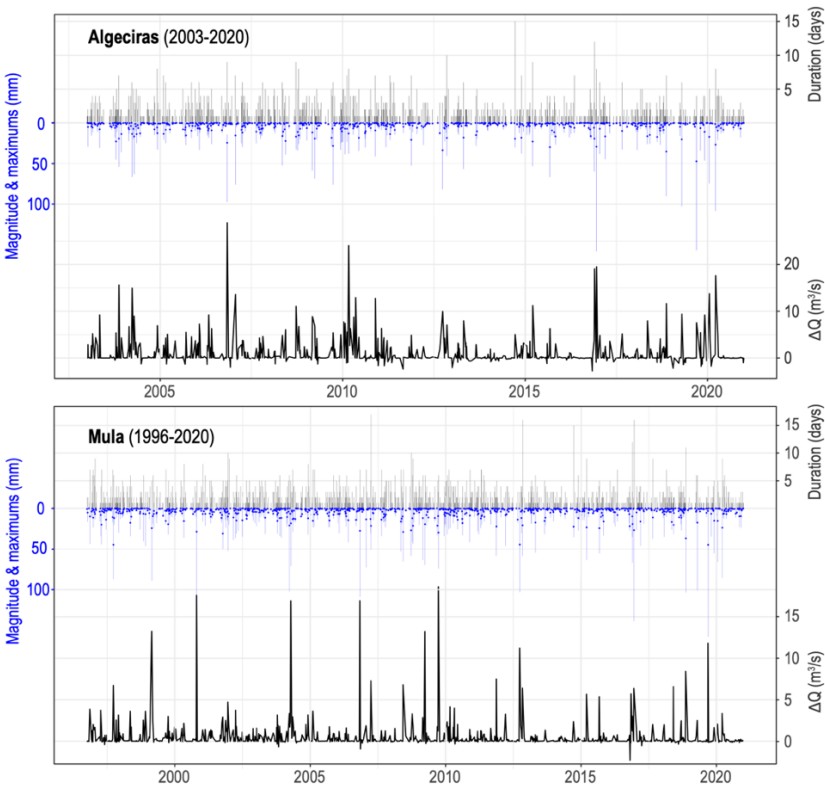

Figure 5: Rainfall events (*RE*) in Algeciras (upper row) and Mula (lower row) showing the magnitude of the *RE* (blue bars), the sum of hourly maximums (blue dots), the duration of the *RE* (narrow black bars over magnitudes) and the flow contributed by the *RE* (thick continuous black lines).

Table 2: Monthly percentage of non-contributing RE (rainfall events producing zero or negative *ΔQ*).

|  | J | F | M | A | M | J | J | A | S | O | N | D |
|---|---|---|---|---|---|---|---|---|---|---|---|---|
| **Algeciras** | 38.3 | 37.0 | 32.8 | 26.8 | 36.9 | 46.3 | 58.6 | 44.0 | 35.5 | 38.7 | 24.6 | 34.4 |
| **Mula** | 29.7 | 23.9 | 25.0 | 22.4 | 27.1 | 33.8 | 40.0 | 33.8 | 22.4 | 30.1 | 17.9 | 26.3 |

## 3.2 Linear rainfall-runoff relationships

The linear correlation between the parameters of the RE and their corresponding *ΔQ* showed the general agreement between precipitation and flow contribution. As expected, the parameters derived from the RE, duration, magnitude and hourly maximums were highly positively correlated (Figure 6). An increase in the duration of the events usually led to higher magnitudes of cumulated precipitation (Pearson 0.75 and 0.74 in Algeciras and Mula, respectively), but was the relationship

between magnitudes and cumulated hourly maximums the most direct with Pearson correlations of 0.98. These positive relationships between the parameters, which are almost identical in both watersheds, showed that the majority of the events are torrential (hourly maximums represent a higher proportion of the magnitudes) and of short duration (most of them occur between 1 and 5 days). However, the relationship between the RE parameters and $\Delta Q$ was very similar between watersheds. Both showed positive correlations, Algeciras revealed values from 0.63 to 0.73, with a more direct response to the duration of RE and a slightly lower, and very similar, to the magnitude and maximums. In a lesser intensity, Mula showed a similar overall pattern but with slightly higher Pearson value in relation to duration of the events (0.69). These results indicated that the flow reaction to the RE was different between both watersheds in terms of the intensity of the relationship and that the linear association is not enough to derive conclusions about it.

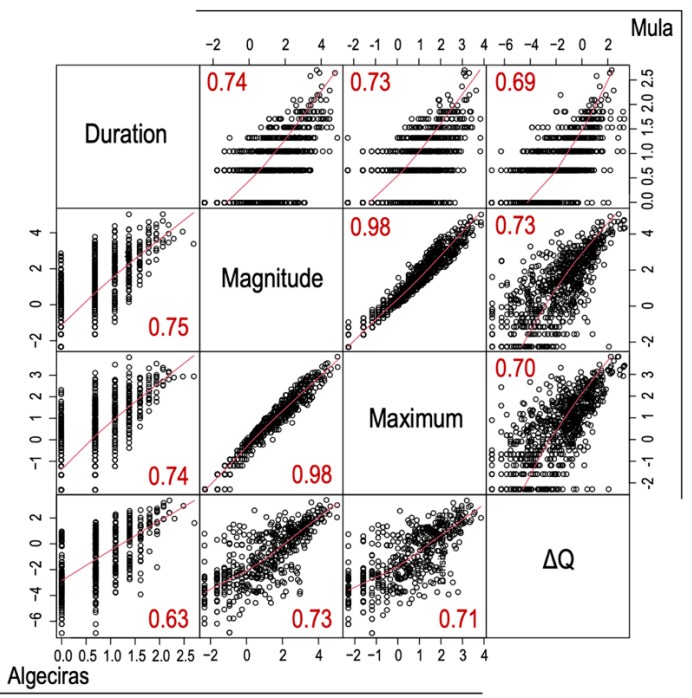

**Figure 6: Values of precipitation variables and flow contribution ($\Delta Q$) of all events in Algeciras (bottom left side) and Mula (top right side). Magnitude and maximum variables are in logarithmic scale. Pearson correlations are shown in red (all correlations are significant at $\alpha < 0.01$)**

**3.3 Nonlinear rainfall-runoff relationships**

Results (Table A1) showed that the model with duration and magnitude (M04) of RE got the lowest AIC in Algeciras. Despite the rest of the estimate errors were not the lowest, M04 was the best combination in which all predictors were significant. Mula watershed showed a similar behaviour but in this case the combination of duration and the cumulated hourly maximums (M05) got the best values with all their predictors significant. Duration was revealed as the key variable

for both watersheds and the total amount of precipitation was more important in Algeciras than in Mula, where the intensity of the RE (maximums) played a fundamental role on the flow generation. GAM models were finally calculated with duration and magnitude for Algeciras and with duration and cumulated hourly maximums for Mula (Table 3).

**Table 3: GAM summaries for both watersheds.**

| Algeciras | | | | |
|---|---|---|---|---|
| **Parametric coefficients:** | | | | |
| | **Estimate** | **Std. Error** | **z value** | **Pr(>\|z\|)** |
| **(Intercept)** | 1.998 | 1.517 | 1.317 | 0.188 |
| **Approximate significance of smooth terms:** | | | | |
| | **edf** | **Ref.df** | **Chi.sq** | **p-value** |
| **s(duration)** | 2.908 | 3.106 | 40.64 | < 2e-16 |
| **s(magnitude)** | 3.385 | 4.025 | 28.33 | 1.17e-05 |
| **R-sq.(adj) = 0.312** | **Dev. expl. = 28.7%** | **UBRE = -0.045623** | **n = 720** | |
| Upper Mula | | | | |
| **Parametric coefficients:** | | | | |
| | **Estimate** | **Std. Error** | **z value** | **Pr(>\|z\|)** |
| **(Intercept)** | 3.174 | 2.123 | 1.496 | 0.135 |
| **Approximate significance of smooth terms:** | | | | |
| | **edf** | **Ref.df** | **Chi.sq** | **p-value** |
| **s(duration)** | 3.302 | 3.599 | 108.55 | <2e-16 |
| **s(maximum)** | 2.042 | 2.495 | 10.27 | 0.0108 |
| **R-sq.(adj) = 0.312** | **Dev. expl. = 30.5%** | **UBRE = -0.17734** | **n = 985** | |

Contingency table (Table 4) showed a general success rate (positive and negative) of 75.97% in Algeciras and 77.77% in Mula. True positives were 76.3 and 77.9% for Algeciras and Mula, respectively, representing the correctly predicted RE with flow generation. False negatives (wrongly predicted *Qbin*) were 24.5 and 22.6% of the cases. True negatives, indicating the correctly predicted non-contributing RE were 75.5 and 77.4% and false positives (wrongly predicted contributing RE) were 23.7 and 22.1%.

While success rates are relatively high in both watersheds, results suggest other variables driving flow generation in RE different than precipitation. Again, topographical and soil characteristics, as well as other climatic factors such as evaporation, probably play an important role that is difficult to integrate in these types of models.

**Table 4: Contingency table of observed (Obs) and predicted (Pred) *Qbin* for Algeciras (regular text) and Mula (italic text) with number of cases and percentage (in brackets) of true and false positives and negatives.**

| | **Obs = 0** | **Obs = 1** |
|---|---|---|
| **Pred = 0** | 197 (75.5%) *205 (77.4%)* | 64 (24.5%) *60 (22.6%)* |
| **Pred = 1** | 109 (23.7%) *159 (22.1%)* | 350 (76.3%) *561 (77.9%)* |

70 Diagnostic plots of the partial effects (Figure 7) showed the probability of flow generation by a RE as long as the rest of the partial effects remain in their average values. For instance, Algeciras showed that an event of 5 days duration guarantees the flow contribution at a 95% probability (Figure 7a), but the 2-day RE already sum a probability of 50%. On the other hand, in a RE of average duration (1.9 days), the magnitude required to reach 95% probability of flow contribution is 20.7 mm (heavy rainfall), but the 50% probability is reached (Figure 7b) with 0.1 mm, meaning any precipitation record. The

75 maximum probability of flow contribution is 99.5% with 158.3 mm. By comparison, Mula requires a 4-day RE to ensure new flow generation with a 95% probability. However, considering an average duration event (2.1 days), the cumulated hourly maximums needed to fulfil with that probability is 3.8 (not very intense precipitation), being reduced to 0.1 for a 50% probability.

Overall, these results indicate that, despite the new flow generation similarly reacts to RE in Algeciras and Mula, in both

80 watersheds the duration of the event is a critical factor. However, the total amount of precipitation is more important in Algeciras than Mula, where cumulated hourly maximums, ultimately, the intensity of the RE has a more direct relationship.

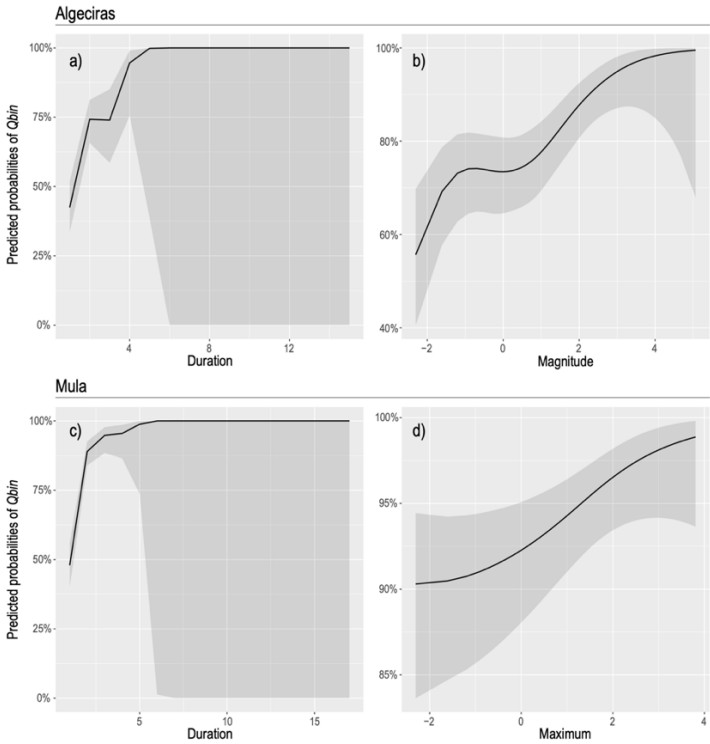

**Figure 7: Predicted probabilities of partial effects of individual smooths for Algeciras (a, b) and Mula (c, d). Shadowed areas show the 95% confidence intervals. Magnitudes and maximums are in logarithmic scale.**

### 3.4 Return periods of RE

We calculated the return levels of magnitude of the RE in Algeciras and of cumulated hourly maximums in Mula for different return periods (Figure 8). We used the POT values of RE exceeding a particular threshold (see Figure A1 and A2 for threshold selection) to adjust them to a GPD. Thresholds were 25 mm for Algeciras and 7 mm for Mula that, based on the GAM models, represent the 95.9% and 96.4% probabilities of flow generation, respectively.These thresholds mean that all RE in Algeciras with magnitudes lower than 25 mm, and all REin Mula with cumulated hourly maximums lower than 7 mm,can occur every year and, therefore, the probability of flow generation at 95% in both watersheds has a return period lower than 1 year. However, the RE ensuring the flow generation at a probability higher than 98% span return periods from 2 to >100 years.This large difference in the return periodsreveals the extreme irregularity of flows in Mula and the high uncertainty in prediction based only on the RE.

The maximum probability of flow generation that the GAM model was able to predict for Algeciras, being the duration in its average value (1.9 days), was 99.5%, which corresponds with a RE of magnitude of 158.3 mm (sum of total precipitation). According to the fitted POT values to a GPD, the return period of this magnitude ranged from 15to 30 years. However, this return period is dramatically reduced with low flow generation probabilities, meaning that high-magnitude episodes (e.g., higher than 150 mm) are rare but of key importance to ensure flow generation.Similar results were obtained for Mula, where the maximum probability(98.8%) of flow generationimplied an RE with a cumulated hourly maximum of 44.6 mm, which represents a return period near to 50 years.

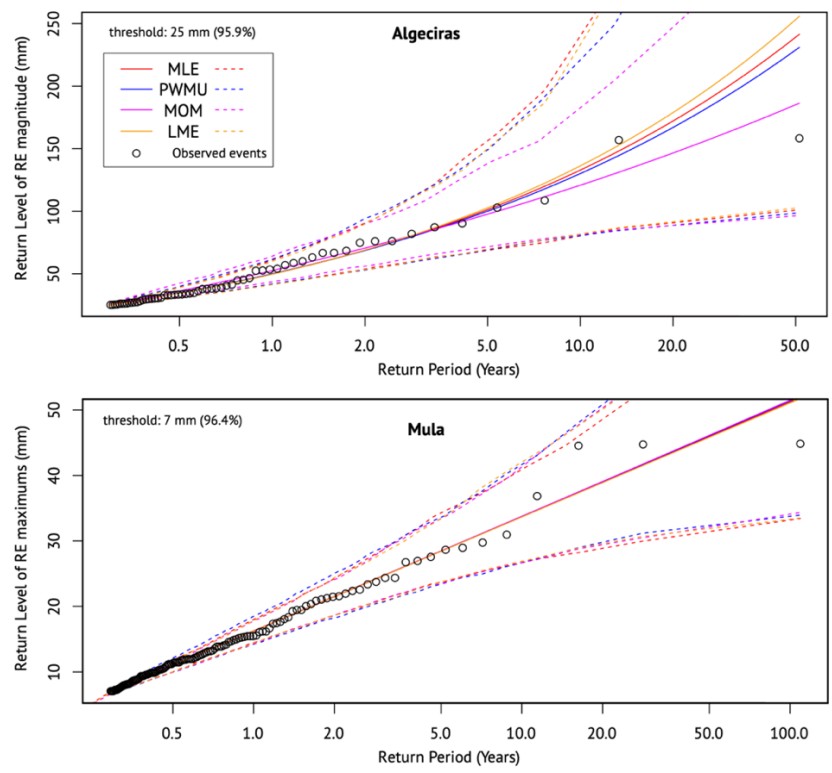

**Figure 8: Return levels (RL) of magnitude of the events in Algeciras (top) and cumulated hourly maximums in Mula (bottom). Solid lines show the RL estimated for different return periods with four different methods: Maximum Likelihood Estimation (MLE); Unbiased Probability Weighted Moments (PWMU); Moments (MOM); and Likelihood Moment (LME). Dashed lines show the confidence intervals. Dots are the observed magnitude and maximums of Algeciras and Mula, respectively. RL of 98% and maximum probabilities of flow generation are indicated.**

## 4 Discussion

Rainfall-runoff relationships at event scale in Upper Mula and Algeciras showed very different flowdynamics.Although they are located near each other and precipitation regimesare relatively similar, the response to RE in terms of flow generation had in common the responsibility of the duration of the event but the magnitude and the intensity played a different role depending on the watershed (Figure 7).Differences in the lithological setting also explain these dissimilarities, agreeing with previous worksin similar environments (e.g., Huza et al., 2014; Merheb et al., 2016; Fortesa et al., 2020; Martinez-Salvador and Conesa-García, 2020). Constrained to the study area of our research, Martínez-Salvador et al. (2021) noted that flows in Upper Mula sourcefrom lateral flow and from base flow storage, due to the permeable materials.Conversely,the ephemeral stream in Algeciras is caused by the low values of the saturated hydraulic conductivity, the hydraulic conductivity of the main channel, and the coefficient of roughness for overland flow, since a large part of the basin is dominated by clayey materials, emphasizingthe importance of lateral flow within the kinematic storage model.Thus, in addition to the dependence

on the lithological and terrain configuration (van Dijk, 2010) and changes in seasonal precipitation regimes (Fakir et al., 2021), the RE duration, intensity, and magnitude, have a high probability of changing the available flow, as shown in the results of the GAM model. For instance, Camarasa (2021) showed that runoff in ephemeral streams is more dependent on rainfall intensity in the Mediterranean area than in non-arid environments, and Gutiérrez-Jurado et al. (2019) and Bull et al. (2000) showedthat soil type has the greatest influence on flow generation in intermittent rivers.In summary, rainfall-runoff relationships in ephemeral streams are influenced by topography and soil characteristics (Wooldridge et al., 2003; Chen et al., 2019), however,their flows are heavily dependent on the intensity, which is usually considered as the ratio between the volume of rainfall (magnitude) in a RE and its duration(e.g., Camarasa and Tilford, 2002; La Torre Torres et al., 2011;El Afy, 2016). In addition to the topographical and climatic characteristics of the watersheds, anthropic interventions such as irrigation, industrial uses,roads, or any water resources change at large scale, can modify rainfall-runoff dynamics, leading to increased consequences of flooding (Conesa-García et al., 2016; Betancourt-Suárez et al., 2021).

Most of the previous works based on rainfall-runoff modelling in ephemeral streams were dedicated to runoff forecasting based on rainfall and topographical characteristics at different temporal and spatial scales. Many of these studies used different methods such as transfer-function models (Camarasa et al., 2002), artificial neural networks (Daliakopoulos and Tsanis, 2016; Ahmadi et al., 2019), or hydraulic models (Berardi et al., 2013; Doglioni et al., 2015), amongst others. While they fall into the categories of conceptual or physics-based models (Wheater et al., 1993), our focus is a metric approach using rainfall observations at event scale to characterize the response of flow generation. To this end, we used a GAM method instead of other regression procedures because of its ability to handle nonlinear relationships between the response variable (flow generation) and the set of explanatory variables (Paillex et al., 2019). GAM models have been already used to model rainfall-runoff relationships in ephemeral streams (e.g., van Ogtrop et al., 2011; García-Galiano et al., 2015; Rashid and Beechman, 2019), and they are highly appropriate for these semi-arid environments since they involve the usual highly nonlinear relationships between rainfall and runoff in this type of intermittent rivers (Ye et al., 1997; Goodrich et al., 1997). However, the novelty of our research is vested in the use of the characteristics of rainfall events (duration, magnitude, and maximums) as explanatory variables, instead of the conventional analysis using all rainfall observations (daily, monthly, or annual) without our proposed distinction. Our approach allows to separate the rainfall-runoff responses by the occurrence of rainfall events (consecutive rainy days), avoiding inconsistencies in flow generation of consecutive rainy days due to potential lags between rainfall at headwaters and flow at gaugesin lowlands. While the event scale is not new in ephemeral streams studies, most of the event-based analyses are referred to experimental designs based on single or a few events, and/or in sub-daily scales (e.g., De Boer, 1992; Bull et al., 2000; Gutierrez-Jurado et al., 2019). By isolating the rainfall events from daily data over a long period, we provide a general overview of the response of runoff to rainfall. The selection of the explanatory variables was based on the core characteristics of a RE: duration, magnitude (sum of precipitation in the total duration of the event), and intensity (through the sum of hourly maximums). These three variables have been widely used in rainfall-runoff modelling of ephemeral streams (e.g., Camarasa et al., 2002; Kirkby et al., 2005; Hooke, 2016) and represent the rainfall characteristics influencing on runoff generation (Martínez-Mena et al., 1998; Ran et al., 2012; dos

Santos, 2017).The atmospheric evaporative demand measured in terms of reference evapotranspiration is well known to be a useful climatic factor modelling runoff (Gallart et al., 2002; Goulden and Bales, 2014; Roy et al., 2017).However, we did not use it in our analysis because we pursued unravelling the particular contribution of rainfall, at event scale, on the runoff generation only using precipitation observations to create a reliable model representing that contribution.

Precipitation behaviour over the last decades in both watersheds was slightly different than the rest of the Iberian Peninsula, where a decrease in the intensity prevailed (Serrano-Notivoli et al., 2018). However, the Mediterranean Spanish coast, and especially the southeast area where Algeciras and Upper Mula are located, experimented a moderate increase of high and very high precipitation events from mid-20[th] century as well as a remarkable increase in the number of wet days, agreeing with temporal patterns of both watersheds (Figure 3). While the precipitation totals decrease is an already well-known trend (Gonzalez-Hidalgo et al., 2011; Homar et al., 2010; Ruiz-Sinoga et al., 2010), southeastern Spain tended to a more intense precipitation (Mosmann et al., 2004) and more concentrated in time (De Luis et al., 2011; Serrano-Notivoli et al., 2017c). This scenario increases the chances of flow generation in ephemeral streams of Algeciras and Mula, but the high irregularity and the negative trend of precipitation totals do not envisage a significant change on flow dynamics to less intermittent streams. However, a change in the seasonality of flows is expected under these changing conditions of precipitation, leading to potential alterations that could intensify wet and dry periods (Pumo et al., 2016). In Algeciras and Upper Mula watersheds, climate change scenarios also depict a decrease in water resources caused by the changing seasonality, due to an increased evapotranspiration situation (Martínez-Salvador, et al., 2021).

Linear rainfall-runoff relationships were clearly uninformative due to the great irregularity of the RE and they did not provide a valid approach to derive rainfall thresholds (T) for flow generation. For this reason, we used a GAM approach, that takes advantage of non-linear relationships, which are highly representative of the great irregularity of precipitation in the Mediterranean area. This approach represents an advantage among the wide variety of methods that has been previously used to model these thresholds in ephemeral or low-yield streams such as multivariate regressions, machine learning approaches, etc. (e.g., Kaplan et al., 2020; Kampf et al., 2018; Shortridge et al., 2016). Furthermore, GAMs allow for avoiding stationarity assumptions in rainfall-runoff relationships (Tian et al., 2020)in comparison with the abovementioned methods. Using nonparametric smoothed functions as a response curve for each variable has been demonstrated to reinforce the capture of non-linearity between dependent variable (*Qbin* in our case) and covariates (RE parameters) in hydrological models (Rahman et al., 2018). However, the accuracy of GAMs models is highly dependent on the data since the predictability is jeopardized when the smoothed variables contain outliers, which is precisely the case of the great variability of the RE parameters. The own nature of GAMs, being accurate in the data range, can lead to overfitting and a loss of predictability in uneven data sets. Yet, obtained rainfall characteristics for Algeciras and Mula are similar to those exposed by Hooke (2016) in a near watershed (Guadalentín basin).

Low return periods were shown for events generating new flow at 95% probability, but they dramatically increased when probabilities were increased until maximum (99.5% in Algeciras and 98.8% in Mula). However, the analysis has some limitations to consider. First, we only considered one variable (magnitude or maximum) for each basin when, in fact, they

also depend on duration. This means that the return periods could be higher because the degree of reliability provided by the model only considers the situation in which those variables occur in a RE of average duration (1.9 and 2.1 days, respectively). In this regard, further investigation is needed to set more accurate return periods because univariate approaches might lead to inadequate estimation of the risk of a RE (Brunner et al., 2016).It should be also considered that we only used the data of the RE in periods when flow was available (18 years for Algeciras and 25 years for Upper Mula) because hourly maximums were not available out of the considered periods, meaning that the obtained return periods could be lower if including long-term data series. Additionally,a non-stationary POT approach would be more appropriate, as made in previous works (e.g.Beguería et al., 2010; Agilan et al., 2021), but longer data series are needed to build reliable fittings of distributions.

Lastly, the nonlinear analysis of RE helped to understand the type of events required to generate new flow in both watersheds. Prediction models in hydrology are a useful tool to improve water resources management in ephemeral streams through a deeper knowledge of their rainfall-runoff dynamics, especially in vulnerable areas to thepotential effects of climate change and the accelerated degradation of their ecosystems.

## 5 Conclusions

We analysed rainfall-runoff relationships of two intermittent streams located in two medium-sized watersheds in southern mainland Spain: Algeciras (2003-2020) and Upper Mula (1996-2020), with the aim of modelling the type of rainfall event required to generate new flow. While a linear relationship was insufficient to derive robust conclusions about flow production and rainfall, a nonlinear analysis using GAMs helped to understand that most of the new flow is driven by a similar duration of the rainfall events (4-5 days to ensure a 95% probability) in both watersheds. However, the magnitude of the event (cumulated precipitation) was a more significant predictor in Algeciras (20.7 mm) than Upper Mula, where cumulated hourly maximums of each day (3.8 mm) showed a higher significance than in Algeciras. These differences could be due to the different orographic and lithological configuration. For example, Algeciras is smaller, with a higher average slope than Upper Mula and less permeable materials prevailing across the watershed, in comparison to Upper Mula where groundwater plays an important role on water management from rainfall events and producing a different response than Algeciras.

Results showed that the precipitation regime was very irregular, and the observed average event of 1.2 days and less than 1.5 mm was clearly insufficient to generate new flow. Almost a third part of the rainfall events were non-contributing for flow generation (flows were similar or lower than previous day to the rainfall event). A long-term analysis through the calculation of return levels showed that lowrainfall return periods are enough to produce a contributing rainfall event with a 95%, rapidly increasing when raising flow generation probabilities. These results agree with the long-term (70 years) precipitation patterns, that showed a highly variable annual water availability alongside a significant increase of wet days, with different behaviour among watersheds. Within the study period, Upper Mula showed 16 of 25 years below average precipitation,

while Algeciras remained with the same frequency as previous decades but a higher rate of wet days. A future drier scenario as considered in western Mediterranean climate projections could lead to increase the return periods for the required magnitude of rainfall events to generate flows.

## Data availability

Daily and hourly precipitation data belong to different institutions in Spain (see section 2) and can be accessed through formal requests. The SPREAD gridded daily precipitation dataset is described and provided in Serrano-Notivoli et al. (2017). Daily flow data series source from CEDEX (https://ceh.cedex.es/anuarioaforos/default.asp).Data series of rainfall events for both watersheds used for the statistical analysis have been published in open access: https://doi.org/10.5281/zenodo.5801008.

## Author contribution

RSN and CCG developed the research idea and were responsible for conceptualization. RSN, AMS, RGL and DES processed the data, designed the visualizations, and validated results. RSN developed the statistical analysis and prepared the manuscript with the contribution from all the co-authors.

## Competing interests

The authors declare that they have no conflict of interest.

## Acknowledgments

This work has been financed by ERDF (FEDER) funds / Spanish Ministry of Science, Innovation and Universities - State Research Agency (AEI) / Project CGL2017-84625-C2-1-R (CCAMICEM); State Program for Research, Development and Innovation oriented to the Challenges of Society. We also would like to extend our thanks to the State Meteorology Agency (AEMET), in Spain, for providing meteorological data, and to the Segura River Hydrographic Confederation Center (SHC), Government of Spain, for its collaboration. RSN is supported by the Government of Aragón through the "Program of research groups" (group H09_20R, "Climate, Water, Global Change, and Natural Systems").

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

**Appendix A**

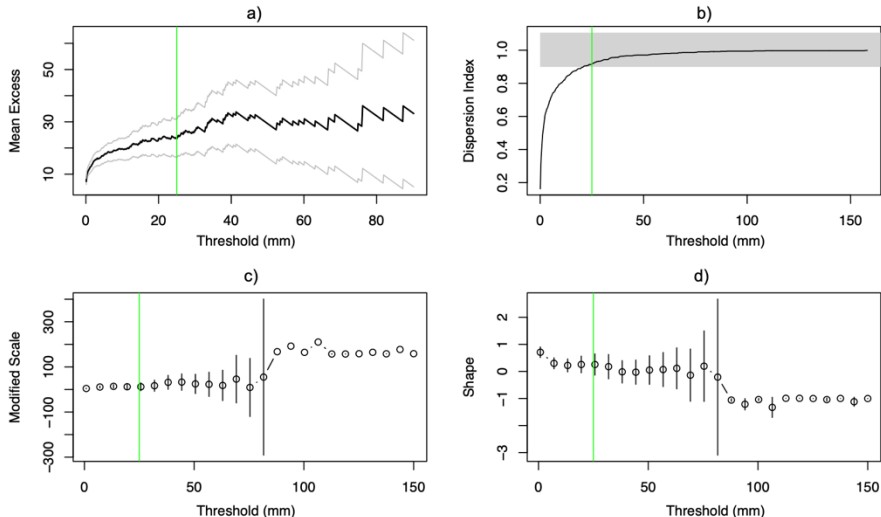

**FigureA1: Graphical summary of RE threshold ($\mu$) selection in Algeciras: a) Mean Residual Life: mean value of observations over a threshold (mean excess); b) Dispersion Index; c) and d) scale and shape parameters estimates from the GPD for a range of values of $\mu$. Green line represents the $\mu$(25 mm) selected, implying a higher variability of its exceeding values in a), c) and d), and posing a limit in b) from which dispersion index estimates are near the theoretical value 1.**

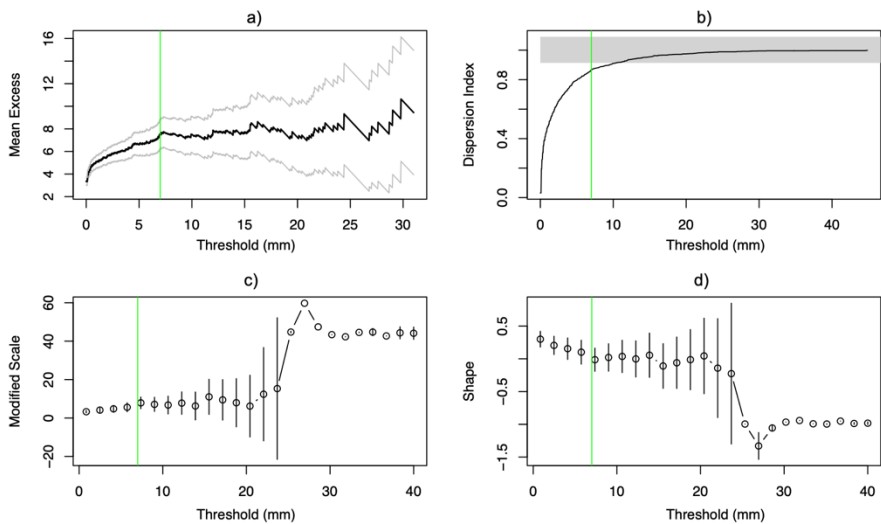


    **FigureA2: Graphical summary of RE threshold ($\mu$) selection in Upper Mula: a) Mean Residual Life: mean value of observations over a threshold (mean excess); b) Dispersion Index; c) and d) scale and shape parameters estimates from the GPD for a range of values of $\mu$. Green line represents the $\mu$ (7 mm) selected, implying a higher variability of its exceeding values in a), c) and d), and posing a limit in b) from which dispersion index estimates are near the theoretical value 1.**


**Table A1: Accuracy assessment of the models for Algeciras (regular text) and Upper Mula (italic text). Goodness-of-fit measures: AIC (Akaike information criterion), logLiK (log-likelihood), deviance (Residual deviance), UBRE (Un-Biased Risk Estimator) and number of significant predictors. Bold text indicates the values of the selected model.**

| Model | Variables | AIC | logLik | deviance | UBRE | Signif. preds. |
|-------|-----------|-----|--------|----------|------|-------|
| M01 | Duration | 715.955 | -354.058 | 708.117 | -0.00562 | 1/1 |
|     |          | *818.246* | *-404.761* | *809.521* | *-0.16929* | *1/1* |
| M02 | Magnitude | 738.640 | -364.67 | 729.341 | 0.02589 | 1/1 |
|     |           | *939.895* | *-467.102* | *934.203* | *-0.04579* | *1/1* |
| M03 | Maximum | 755.445 | -373.294 | 746.589 | 0.04923 | 1/1 |
|     |         | *944.966* | *-467.762* | *935.524* | *-0.04064* | *1/1* |
| M04 | Duration+Magnitude | **687.151** | **-336.282** | **672.564** | **-0.04562** | **2/2** |
|     |                    | *811.434* | *-399.792* | *799.584* | *-0.17621* | *2/2* |
| M05 | Duration+Maximum | 694.739 | -340.1 | 680.2 | -0.03508 | 2/2 |
|     |                  | ***810.325*** | ***-398.818*** | ***797.636*** | ***-0.17734*** | ***2/2*** |
| M06 | Magnitude+Maximum | 688.426 | -363.667 | 727.334 | 0.02761 | 1/2 |
|     |                   | *940.335* | *-464.357* | *928.713* | *-0.04535* | *1/2* |
| M07 | Duration+ Magnitude+Maximum | 688.426 | -335.622 | 671.244 | -0.04385 | 2/3 |
|     |                             | *812.278* | *-398.779* | *797.559* | *-0.17535* | *1/3* |

**Table A2: Concurvity between smooth functions of the predictors in the GAM analysing flow contribution by the RE (*Qbin*) for Algeciras (regular text) and Mula (italic text). Zero means no concurvity among covariates, one means complete concurvity.**

|          | para | s(duration) s(duration) | s(magnitude) s(maximum) |
|----------|------|-------------------------|-------------------------|
| **worst** | 0 | 0.59 | 0.59 |
|           | *0* | *0.55* | *0.55* |
| **observed** | 0 | 0.39 | 0.57 |
|              | *0* | *0.33* | *0.53* |
| **estimate** | 0 | 0.38 | 0.22 |
|              | *0* | *0.37* | *0.22* |
