# Peer review of "Rainfall-runoff relationships at event scale in western Mediterranean ephemeral streams"

_Hydrology and Earth System Sciences, 2021_

## Referee Comment (RC2)

[referee-annotated manuscript omitted]

---

## Author Comment (AC1)

**RESPONSE TO REVIEWER**

**Review of Manuscript No.: hess-2021-352**

**Title: "Rainfall-runoff relationships at event scale in western Mediterranean ephemeral streams**

**Authors:** Roberto Serrano-Notivoli, Alberto Martínez-Salvador, Rafael García-Lorenzo, David Espín-Sánchez, and Carmelo Conesa-García

We are grateful to the Reviewer for his/her thoughtful and constructive feedback. In this Response to the Reviewer's file, we provide complete documentation of the changes that have been made in response to the reviewer's suggestions and comments. The original comments are shown in **bold text** and the author responses are shown indented in plain text. Quotations from the revised manuscript are shown in *italic text*. Line numbers in the author responses refer to locations in the revised manuscript.

**Referee #1**

**This paper aims at better understanding rainfall-runoff relationships through statistical modelling in two ephemeral streams in Spain (with a focus on rainfall events triggering runoff). The paper is well structured. The objectives are also clearly presented.**

**Evapotranspiration is probably another driver (see L180) – depending when extreme events occur, response in terms of runoff may differ with the stage of plant growth. Why have you not introduced ET0 data (e.g. https://essd.copernicus.org/articles/11/1917/2019/) in your analyses? e.g. considering P- ET0 as explanatory variable.**

> Thank you for your suggestion. We, indeed, considered the inclusion of ET0 which we are sure that could work as one of the main drivers, especially in summertime as mentioned all along the manuscript. However, the suggested dataset does not fit with our approach of rainfall events isolation. The SPETO dataset is at a weekly temporal resolution that considers the division of a month in 4 periods, always starting in day 1 and aggregating the last days (29/30/31) in the 4th week depending on the month. This approach avoids combined weeks among consecutive months. The weekly aggregation, although useful for climatic analysis, is not applicable to our study, where we aggregate rainfall events based on daily precipitation data. Additionally, the dataset ends in 2014, meaning that the last 6 years of our period of analysis are not available.

**I have some doubts about the method used for the frequency analysis: obviously, all the episodes have been kept (more than one value sampled each year) and the peak over threshold approach should be carried out to derive return levels. The generalized Pareto distribution is the most suited distribution (instead of GEV adapted for the block maxima method). For example, the empirical return period of the observed maximum and the length of the time series should be in the same order while Figure 8 suggests return periods > 100 years. Consequently, the rainfall events triggering runoff are probably more frequent than those derived from the frequency analysis. The authors have applied the block maxima approach to data resulting from the selection of over-threshold values (threshold = 0). The method and the discussion should both be revised.**

> Thank you for your useful comments. Your argument is right, and we have changed the method to calculate return periods through a *peak-over-threshold* approach. As we now state in methods section (L143-150), this is the most suitable approach due to continuous series of rainfall events are available for both watersheds:

[revised manuscript text omitted]

**There are many studies on rainfall-runoff relationships in ephemeral streams. The authors should develop more the peculiarities of their findings for the two catchments regarding these relationships.**

Thank you for your suggestion. We included several more references to improve the discussion of the results (L359-363; L373-376)

*"[...] For instance, Camarasa (2021) showed that runoff is more dependent on rainfall intensity in the Mediterranean area, and Gutiérrez-Jurado et al. (2019) demonstrated that soil type has the greatest influence on flow generation, as well as Bull et al. (2000) mentioned in a study of a watershed near to our study area. In addition, anthropic interventions such as irrigation, industrial uses, roads, or any water resources change at large scale, can modify rainfall-runoff dynamics, leading to increased consequences of flooding (Conesa-García et al., 2016; Betancourt-Suárez et al., 2021)."*

*"However, a change in the seasonality of flows is expected under these changing conditions of precipitation, leading to potential alterations that could intensify wet and dry periods (Pumo et al., 2016). In Algeciras and Upper Mula watersheds, climate change scenarios also depict a decrease in water resources caused by the changing seasonality, due to an increased evapotranspiration situation (Martínez-Salvador, et al., 2021)."*

*Betancourt-Suárez, V., García-Botella, E., Ramón-Morte, A.: Flood mapping proposal in small watersheds: A case study of the rebollos and miranda ephemeral streams (cartagena, Spain). Water, 13(1), 102, https://doi.org/10.3390/w13010102, 2021.*

*Bull, L.J., Kirkby, M.J., Shannon, J., Hooke, J.M.: The impact of rainstorms on floods in ephemeral channels in southeast Spain. Catena, 38(3), 191-209, https://doi.org/10.1016/S0341-8162(99)00071-5, 2000.*

*Camarasa, A.: Flash-flooding of ephemeral streams in the context of climate change. Geog. Res. Lett., 47(1), 121-142, https://doi.org/10.18172/cig.4838, 2021.*

*Conesa-García, C., García-Lorenzo, R., Pérez-Cutillas, P.: Flood hazards at ford stream crossings on ephemeral channels (south-east coast of Spain). Hydrol. Process., 31(3), 731-749, https://doi.org/10.1002/hyp.11082, 2016.*

*Gutiérrez-Jurado, K.Y., Partington, D., Batelaan, O., Cook, P., Shanafield, M.: What Triggers Streamflow for Intermittent Rivers and Ephemeral Streams in Low-Gradient Catchments in Mediterranean Climates. Water Resour. Res., 55(11), 9926-9946, https://doi.org/10.1029/2019WR02504, 2019.*

*Pumo, D., Caracciolo, D., Viola, F., Noto, L.V.: Climate change effects on the hydrological regime of small non-perennial river basins. Sci. Total Environ., 512(A), 76-92, https://doi.org/10.1016/j.scitotenv.2015.10.109, 2016.*

**Details:**

**L35: There is an inversion between first name and last name in the reference « Thibault et al. 2017 ». = = > Datry et al. is the correct reference.**

Modified as suggested.

**L40: a reference regarding sediment transport: https://doi.org/10.1016/j.catena.2020.104865**

Reference added.

**Fig. 1: we do not see the main river network. Please add the location of the two reservoirs, even if we guess that they are the mouths of the two catchments and point out the stations used to compute the precipitation time series.**

Modified as suggested.

**L102-106: The authors used long time series to perform a stationarity analysis. Are gridded and local data consistent during the concomitant period (correlation, mean, etc.)? This is important to assess the representativeness of the gridded data for the two catchments.**

The SPREAD dataset, referenced work as Serrano-Notivoli et al. (2017), spans the period from 1950 to 2012. It was extended until 2020 in the study area using the same data series as used in the rest of the analysis, through the method described in Serrano-Notivoli et al. (2017b) to ensure the reliability of the data. We have added this reference to make clear this point in the methodological section (L110-112).

*"[...] we used the SPREAD dataset (Serrano-Notivoli et al., 2017), a daily gridded precipitation dataset covering the whole Spanish territory at a 5x5 km spatial resolution, to analyse long-term trends of annual precipitation of the two watersheds by extending its period coverage until 2020 following Serrano-Notivoli et al. (2017b)."*

*Serrano-Notivoli, R., de Luis, M. and Beguería, S.: An R package for daily precipitation climate series reconstruction. Environ. Modell. Softw., 89, 190-195, http://dx.doi.org/10.1016/j.envsoft.2016.11.005, 2017b.*

**L218-219 & S2: Some criteria have been computed, but not commented (please add some comments or delete the values).**

Thank you for your comments. We moved the table to supplementary material and referenced in the text the table with the GAM summaries for both watersheds.

**Figs 6, 7 and 8: Please use semi-log plots with the y-axis on a logarithmic scale to make the reading easier.**

Thank you for your suggestion. Figure 6 has been changed to show all variables in logarithmic scale. As this action increased some Pearson values, corresponding texts in the manuscript have been adapted to the new results. Figure 7 and (new figure) 8 are already in a semi-log scale.

[Figure]

*Figure 6: Values of precipitation variables and flow contribution (ΔQ) of all events in Algeciras (bottom left side) and Mula (top right side). Magnitude and maximum variables are in logarithmic scale. Pearson correlations are shown in red (all correlations are significant at α<0.01)*

---

## Author Comment (AC2)

**RESPONSE TO REVIEWER**

**Review of Manuscript No.: hess-2021-352**

**Title: "Rainfall-runoff relationships at event scale in western Mediterranean ephemeral streams**

**Authors:** Roberto Serrano-Notivoli, Alberto Martínez-Salvador, Rafael García-Lorenzo, David Espín-Sánchez, and Carmelo Conesa-García

We are grateful to the Reviewer for his/her thoughtful and constructive feedback. In this Response to the Reviewer's file, we provide complete documentation of the changes that have been made in response to the reviewer's suggestions and comments. The original comments are shown in **bold text** and the author responses are shown indented in plain text. Quotations from the revised manuscript are shown in *italic text*. Line numbers in the author responses refer to locations in the revised manuscript.

**Referee #2**

**The authors propose a study that analyze the transformation rainfall-runoff in semi-arid catchments of Southern Spain, where the ephemeral regime of rivers and the climatic stress may lead to hazardous floods or, on the contrary, to dramatic droughts. The study is quite novel and gives significant insight about precipitation depths and return intervals, which may determine water and sediment flows in the channels. The statistical analysis is fine and suitable for the study aims. The results are presented with clearness and synthesis. Although the study is of good quality, I have three suggestions that may improve the MS:**

**- several methodological sentences are reported in the results sections, and this may confuse the reader. I ask the authors to revise both parts.**

> Thank you for your suggestion. We moved the methodological descriptions in results section to methodology (see detailed lines and paragraphs at the end of this document).

**- although literature about the flow regime in ephemeral channels is not abundant, some other cross-comparisons with similar studies may further valorize the study results**

> Thank you. We added several new references to improve the discussion in all the addressed aspects and to compare our work with similar research in nearby areas and in similar terms (L359-363).

> *"[...] For instance, Camarasa (2021) showed that runoff is more dependent on rainfall intensity in the Mediterranean area, and Gutiérrez-Jurado et al. (2019) demonstrated that soil type has the greatest influence on flow generation, as well as Bull et al. (2000) mentioned in a study of a watershed near to our study area. In addition, anthropic interventions such as irrigation, industrial uses, roads, or any water resources change at large scale, can modify rainfall-runoff dynamics, leading to increased consequences of flooding (Conesa-García et al., 2016; Betancourt-Suárez et al., 2021)."*

> *Betancourt-Suárez, V., García-Botella, E., Ramón-Morte, A.: Flood mapping proposal in small watersheds: A case study of the rebollos and miranda ephemeral streams (cartagena, Spain). Water, 13(1), 102, https://doi.org/10.3390/w13010102, 2021.*

> *Bull, L.J., Kirkby, M.J., Shannon, J., Hooke, J.M.: The impact of rainstorms on floods in ephemeral channels in southeast Spain. Catena, 38(3), 191-209, https://doi.org/10.1016/S0341-8162(99)00071-5, 2000.*

> *Camarasa, A.: Flash-flooding of ephemeral streams in the context of climate change. Geog. Res. Lett., 47(1), 121-142, https://doi.org/10.18172/cig.4838, 2021.*

> *Conesa-García, C., García-Lorenzo, R., Pérez-Cutillas, P.: Flood hazards at ford stream crossings on ephemeral channels (south-east coast of Spain). Hydrol. Process., 31(3), 731-749, https://doi.org/10.1002/hyp.11082, 2016.*

> *Gutiérrez-Jurado, K.Y., Partington, D., Batelaan, O., Cook, P., Shanafield, M.: What Triggers Streamflow for Intermittent Rivers and Ephemeral Streams in Low-Gradient Catchments in Mediterranean Climates. Water Resour. Res., 55(11), 9926-9946, https://doi.org/10.1029/2019WR02504, 2019.*

**- some expectations about the future trends of rainfall-runoff transformation under the forecasted climate change scenarios (higher mean temperature and lower precipitation amounts) in the studied area may be reported on the authors' knowledge and experience.**

Regarding future trends, we added a few lines relating the rainfall-runoff potential changes in both watersheds to the expected decrease in precipitation and increase in temperature, leading to a higher evapotranspiration. As stated in previous works, this scenario, depending on the emissions development, has high probabilities of produce changes in seasonality of flows, increasing risks of floods and droughts. Despite we do not explicitly address climate change scenarios in our study, we appreciate your comment and agree that it deserves to be mentioned (L373-376).

*"However, a change in the seasonality of flows is expected under these changing conditions of precipitation, leading to potential alterations that could intensify wet and dry periods (Pumo et al., 2016). In Algeciras and Upper Mula watersheds, climate change scenarios also depict a decrease in water resources caused by the changing seasonality, due to an increased evapotranspiration situation (Martínez-Salvador, et al., 2021)."*

*Martínez-Salvador, A., Millares, A., Eekhout, J.P.C. and Conesa-García, C.: Assessment of Streamflow from EURO-CORDEX Regional Climate Simulations in Semi-Arid Catchments Using the SWAT Model. Sustainability, 13(13), 7120, https://doi.org/10.3390/su13137120, 2021.*

*Pumo, D., Caracciolo, D., Viola, F., Noto, L.V.: Climate change effects on the hydrological regime of small non-perennial river basins. Sci. Total Environ., 512(A), 76-92, https://doi.org/10.1016/j.scitotenv.2015.10.109, 2016.*

**Some other minor suggestions are reported in the commented MS in attachment.**

Thank you, we addressed all your suggestions, point by point:

**L44: Here, I suggest adding shortly the main results of these studies.** Added a short summary of results of those studies (L44-46)

*"These studies highlight that, in the current Spanish Mediterranean scenario of decrease of total amounts of precipitation but increase in intensity (Serrano-Notivoli et al., 2018), hydrological connectivity is more dependent on rain intensity."*

*Serrano-Notivoli, R., Beguería, S., Saz, M.A., de Luis, M.: Recent trends reveal decreasing intensity of daily precipitation in Spain. Int. J. Climatol., 38(11), 4211-4224, https://doi.org/10.1002/joc.5562, 2018.*

**L49: Could you express a research hypothesis?** Thank you for the suggestion. Now, the research hypothesis is stated in a few lines, closing the introduction section. (L51-53)

*"Based on the watershed physical and climatic characteristics, we hypothesise that rainfall-runoff relationships are based in the intensity and magnitude of singular events, strongly dependent on seasonality of precipitation."*

**L93: What do you mean for "reliable"? Please be more specific.** Thank you. We agree that the term can be confusing and we have removed it without changing the meaning of the sentence.

**L96: How do you ensure this reliability?** We replaced *"reliable"* by *"average"*.

**L113: Not clear why you summed the hourly maximums.** We added an explanation to make clear the reason of summing hourly maximums. (L120)

*"to be representative of the amount of precipitation corresponding to the hours of maximum rainfall"*

**L128: Or "return interval"?** We computed the "return levels" for different "recurrence intervals" (or "return periods"). This paragraph was completely rewritten due to a change in the method of frequency analysis proposed by Referee #1.

*To contextualize the different thresholds of the RE for different probabilities of generating flow in both watersheds, we estimated the return levels of the RE using the generalized Pareto distribution (GPD) for extreme events using the peak-over-threshold (POT) approach. POT is most suitable when complete time series (as RE) are available due to all values exceeding a certain threshold can serve as basis for model fitting (Coles, 2001). We used four different estimators to fit the POT data to a GPD (Maximum Likelihood Estimation (MLE); Unbiased Probability Weighted Moments (PWMU); Moments (MOM); and Likelihood Moment (LME)) to establish proper and wide*

*confidence levels in the estimate of maximum rainfall per RE. Thresholds for the asymptotic approximation by a GPD in both watersheds were manually selected through the graphical representation of Mean Residual Life, the Dispersion index and the scale and shape parameters (see Figure S1 and S2).*

**L159: Significantly?** Modified as suggested.

**Figure 4: Better "Study period".** Modified as suggested.

**L171: Please use a more technical term. Perhaps "quicker" or "higher"?** The expression was changed by the term "faster".

**L201: "the majority"** Modified as suggested.

**L215-220: All this part is methodological and should be moved in that section.** Thank you for your suggestion. Indeed, this part better fits in methodological section, and we moved it accordingly.

**L233-234: I ask the authors to reconsider whether this part may be moved to the methodological section.** Thank you. We moved this part to the methodological section.

**Figure 8: Better to reduce the lables on y-axis.** This figure completely changed to show the results based on a different method of frequency analysis calculation based on a suggestion from Referee #1. Now, labels in Y-axis are better readable.

**L299-302: The location of this part may be also reconsidered.** Thank you for your suggestion. Instead of moving this part of the text, which fits in the linear–non-linear comments of the discussion section, we rewrote it to promote a more fluid reading (L379-380).

*"For this reason, we used a GAM approach, that takes advantage of non-linear relationships, which are highly representative of the great irregularity of precipitation in the Mediterranean area. This approach represents an advantage among the wide variety of methods that has been previously used to model these thresholds in ephemeral or low-yield streams such as multivariate regressions, machine learning approaches, etc. (e.g., Kaplan et al., 2020; Kampf et al., 2018; Shortridge et al., 2016). Furthermore, GAMs allow for avoiding stationarity assumptions in rainfall-runoff relationships (Tian et al., 2020) in comparison with the abovementioned methods."*

**L308: Please reconsider the form of this sentence.** Thank you. Based on the new explanations of the POT method now used for frequency analysis, this part of the text has been slightly changed. Now, the sentence is clearer and more informative (L396-398):

*"Low return periods were shown for events generating new flow at 95% probability, but they dramatically increased when probabilities were increased until maximum (99.5% in Algeciras and 98.8% in Mula). However, the analysis has some limitations to consider."*

---

## Editor Decision (ED1)

**RESPONSE TO REVIEWERS**

**Review of Manuscript No.: hess-2021-352**

**Title: "Rainfall-runoff relationships at event scale in western Mediterranean ephemeral streams**

**Authors:** Roberto Serrano-Notivoli, Alberto Martínez-Salvador, Rafael García-Lorenzo, David Espín-Sánchez, and Carmelo Conesa-García

We are grateful to the Reviewers for their thoughtful and constructive feedback, and the Editor for considering a revision. In this Response to the Reviewers' files, we provide complete documentation of the changes that have been made in response to the reviewers' suggestions and comments. The original comments are shown in **bold text** and the author responses are shown indented in plain text. Quotations from the revised manuscript are shown in *italic text*. Line numbers in the author responses refer to locations in the revised manuscript.

**Referee #1**

**This paper aims at better understanding rainfall-runoff relationships through statistical modelling in two ephemeral streams in Spain (with a focus on rainfall events triggering runoff). The paper is well structured. The objectives are also clearly presented.**

**Evapotranspiration is probably another driver (see L180) – depending when extreme events occur, response in terms of runoff may differ with the stage of plant growth. Why have you not introduced ET0 data (e.g. https://essd.copernicus.org/articles/11/1917/2019/) in your analyses? e.g. considering P- ET0 as explanatory variable.**

> Thank you for your suggestion. We, indeed, considered the inclusion of ET0 which we are sure that could work as one of [...]. [...] mentioned all along the manuscript. However, th[...] [...]ach of rainfall events isolation. The SPETO da[...] [...]siders the division of a month in 4 periods, alwa[...] [...]s (29/30/31) in the 4th week depending on the m[...] [...]g consecutive months. The weekly aggregation, a[...] [...]ble to our study, where we aggregate rainfall eve[...] [...]ly, the dataset ends in 2014, meaning that the last 6 years of our period of analysis are not available.

Why do you not have any other data set? Could you at least say why you cannot use ET0?

How do you plant to mention this in the manuscript?

**I have some doubts about the method used for the frequency analysis: obviously, all the episodes have been kept (more than one value sampled each year) and the peak over threshold approach should be carried out to derive return levels. The ==generalized Pareto distribution is the most suited distribution== (instead of GEV adapted for the block maxima method). For example, the empirical return period of the observed maximum and the length of the time series should be in the same order while Figure 8 suggests return periods > 100 years. Consequently, the rainfall events triggering runoff are probably more frequent than those derived from the frequency analysis. The authors have applied the ==block maxima approach to data resulting from the selection of over-threshold values== (threshold = 0). The method and the discussion should both be revised.**

> Thank you for your useful comments. Your argument is right, and we have changed the method to calculate return periods through a *peak-over-threshold* approach. As we now state in methods section (L143-150), this is the most suitable approach due to continuous series of rainfall events are available for both watersheds:

[revised manuscript text omitted]

**There are many studies on rainfall-runoff relationships in ephemeral streams. The authors should develop more the peculiarities of their findings for the two catchments regarding these relationships.**

Thank you for your suggestion. We included several more references to improve the discussion of the results (L359-363; L373-376)

*"[...] For instance, Camarasa (2021) showed that runoff is more dependent on rainfall intensity in the Mediterranean area, and Gutiérrez-Jurado et al. (2019) demonstrated that soil type has the greatest influence on flow generation, as well as Bull et al. (2000) mentioned in a study of a watershed near to our study area. In addition, anthropic interventions such as irrigation, industrial uses, roads, or any water resources change at large scale, can modify rainfall-runoff dynamics, leading to increased consequences of flooding (Conesa-García et al., 2016; Betancourt-Suárez et al., 2021)."*

*"However, a change in the seasonality of flows is expected under these changing conditions of precipitation, leading to potential alterations that could intensify wet and dry periods (Pumo et al., 2016). In Algeciras and Upper Mula watersheds, climate change scenarios also depict a decrease in water resources caused by the changing seasonality, due to an increased evapotranspiration situation (Martínez-Salvador, et al., 2021)."*

*Betancourt-Suárez, V., García-Botella, E., Ramón-Morte, A.: Flood mapping proposal in small watersheds: A case study of the rebollos and miranda ephemeral streams (cartagena, Spain). Water, 13(1), 102, https://doi.org/10.3390/w13010102, 2021.*

*Bull, L.J., Kirkby, M.J., Shannon, J., Hooke, J.M.: The impact of rainstorms on floods in ephemeral channels in southeast Spain. Catena, 38(3), 191-209, https://doi.org/10.1016/S0341-8162(99)00071-5, 2000.*

*Camarasa, A.: Flash-flooding of ephemeral streams in the context of climate change. Geog. Res. Lett., 47(1), 121-142, https://doi.org/10.18172/cig.4838, 2021.*

*Conesa-García, C., García-Lorenzo, R., Pérez-Cutillas, P.: Flood hazards at ford stream crossings on ephemeral channels (south-east coast of Spain). Hydrol. Process., 31(3), 731-749, https://doi.org/10.1002/hyp.11082, 2016.*

*Gutiérrez-Jurado, K.Y., Partington, D., Batelaan, O., Cook, P., Shanafield, M.: What Triggers Streamflow for Intermittent Rivers and Ephemeral Streams in Low-Gradient Catchments in Mediterranean Climates. Water Resour. Res., 55(11), 9926-9946, https://doi.org/10.1029/2019WR02504, 2019.*

*Pumo, D., Caracciolo, D., Viola, F., Noto, L.V.: Climate change effects on the hydrological regime of small non-perennial river basins. Sci. Total Environ., 512(A), 76-92, https://doi.org/10.1016/j.scitotenv.2015.10.109, 2016.*

**Details:**

**L35: There is an inversion between first name and last name in the reference « Thibault et al. 2017 ». = = > Datry et al. is the correct reference.**

Modified as suggested.

**L40: a reference regarding sediment transport: https://doi.org/10.1016/j.catena.2020.104865**

Reference added.

**Fig. 1: we do not see the main river network. Please add the location of the two reservoirs, even if we guess that they are the mouths of the two catchments and point out the stations used to compute the precipitation time series.**

Modified as suggested.

**L102-106: The authors used long time series to perform a stationarity analysis. Are gridded and local data consistent during the concomitant period (correlation, mean, etc.)? This is important to assess the representativeness of the gridded data for the two catchments.**

The SPREAD dataset, referenced work as Serrano-Notivoli et al. (2017), spans the period from 1950 to 2012. It was extended until 2020 in the study area using the same data series as used in the rest of the analysis, through the method described in Serrano-Notivoli et al. (2017b) to ensure

the reliability of the data. We have added this reference to make clear this point in the methodological section (L110-112).

*"[…] we used the SPREAD dataset (Serrano-Notivoli et al., 2017), a daily gridded precipitation dataset covering the whole Spanish territory at a 5x5 km spatial resolution, to analyse long-term trends of annual precipitation of the two watersheds by extending its period coverage until 2020 following Serrano-Notivoli et al. (2017b)."*

*Serrano-Notivoli, R., de Luis, M. and Beguería, S.: An R package for daily precipitation climate series reconstruction. Environ. Modell. Softw., 89, 190-195, http://dx.doi.org/10.1016/j.envsoft.2016.11.005, 2017b.*

**L218-219 & S2: Some criteria have been computed, but not commented (please add some comments or delete the values).**

Thank you for your comments. We moved the table to supplementary material and referenced in the text the table with the GAM summaries for both watersheds.

**Figs 6, 7 and 8: Please use semi-log plots with the y-axis on a logarithmic scale to make the reading easier.**

Thank you for your suggestion. Figure 6 has been changed to show all variables in logarithmic scale. As this action increased some Pearson values, corresponding texts in the manuscript have been adapted to the new results. Figure 7 and (new figure) 8 are already in a semi-log scale.

[Figure]

*Figure 6: Values of precipitation variables and flow contribution (ΔQ) of all events in Algeciras (bottom left side) and Mula (top right side). Magnitude and maximum variables are in logarithmic scale. Pearson correlations are shown in red (all correlations are significant at α<0.01)*
* * *
**Referee #2**

**The authors propose a study that analyze the transformation rainfall-runoff in semi-arid catchments of Southern Spain, where the ephemeral regime of rivers and the climatic stress may lead to hazardous floods or, on the contrary, to dramatic droughts. The study is quite novel and gives significant insight about precipitation depths and return intervals, which may determine water and sediment flows in the channels. The statistical analysis is fine and suitable for the study aims. The results are presented with clearness and synthesis. Although the study is of good quality, I have three suggestions that may improve the MS:**

**- several methodological sentences are reported in the results sections, and this may confuse the reader. I ask the authors to revise both parts.**

> Thank you for your suggestion. We moved the methodological descriptions in results section to methodology (see detailed lines and paragraphs at the end of this document).

**- although literature about the flow regime in ephemeral channels is not abundant, some other cross-comparisons with similar studies may further valorize the study results**

> Thank you. We added several new references to improve the discussion in all the addressed aspects and to compare our work with similar research in nearby areas and in similar terms (L359-363)
>
> *"[...] For instance, Camarasa (2021) showed that runoff is more dependent on rainfall intensity in the Mediterranean area, and Gutiérrez-Jurado et al. (2019) demonstrated that soil type has the greatest influence on flow generation, as well as Bull et al. (2000) mentioned in a study of a watershed near to our study area. In addition, anthropic interventions such as irrigation, industrial uses, roads, or any water resources change at large scale, can modify rainfall-runoff dynamics, leading to increased consequences of flooding (Conesa-García et al., 2016; Betancourt-Suárez et al., 2021)."*
>
> *Betancourt-Suárez, V., García-Botella, E., Ramón-Morte, A.: Flood mapping proposal in small watersheds: A case study of the rebollos and miranda ephemeral streams (cartagena, Spain). Water, 13(1), 102, https://doi.org/10.3390/w13010102, 2021.*
>
> *Bull, L.J., Kirkby, M.J., Shannon, J., Hooke, J.M.: The impact of rainstorms on floods in ephemeral channels in southeast Spain. Catena, 38(3), 191-209, https://doi.org/10.1016/S0341-8162(99)00071-5, 2000.*
>
> *Camarasa, A.: Flash-flooding of ephemeral streams in the context of climate change. Geog. Res. Lett., 47(1), 121-142, https://doi.org/10.18172/cig.4838, 2021.*
>
> *Conesa-García, C., García-Lorenzo, R., Pérez-Cutillas, P.: Flood hazards at ford stream crossings on ephemeral channels (south-east coast of Spain). Hydrol. Process., 31(3), 731-749, https://doi.org/10.1002/hyp.11082, 2016.*
>
> *Gutiérrez-Jurado, K.Y., Partington, D., Batelaan, O., Cook, P., Shanafield, M.: What Triggers Streamflow for Intermittent Rivers and Ephemeral Streams in Low-Gradient Catchments in Mediterranean Climates. Water Resour. Res., 55(11), 9926-9946, https://doi.org/10.1029/2019WR02504, 2019.*

**- some expectations about the future trends of rainfall-runoff transformation under the forecasted climate change scenarios (higher mean temperature and lower precipitation amounts) in the studied area may be reported on the authors' knowledge and experience.**

> Regarding future trends, we added a few lines relating the rainfall-runoff potential changes in both watersheds to the expected decrease in precipitation and increase in temperature, leading to a higher evapotranspiration. As stated in previous works, this scenario, depending on the emissions development, has high probabilities of produce changes in seasonality of flows, increasing risks of floods and droughts. Despite we do not explicitly address climate change scenarios in our study, we appreciate your comment and agree that it deserves to be mentioned (L373-376).
>
> *"However, a change in the seasonality of flows is expected under these changing conditions of precipitation, leading to potential alterations that could intensify wet and dry periods (Pumo et al., 2016). In Algeciras and Upper Mula watersheds, climate change scenarios also depict a decrease in water resources caused by the changing seasonality, due to an increased evapotranspiration situation (Martínez-Salvador, et al., 2021)."*

*Martínez-Salvador, A., Millares, A., Eekhout, J.P.C. and Conesa-García, C.: Assessment of Streamflow from EURO-CORDEX Regional Climate Simulations in Semi-Arid Catchments Using the SWAT Model. Sustainability, 13(13), 7120, https://doi.org/10.3390/su13137120, 2021.*

*Pumo, D., Caracciolo, D., Viola, F., Noto, L.V.: Climate change effects on the hydrological regime of small non-perennial river basins. Sci. Total Environ., 512(A), 76-92, https://doi.org/10.1016/j.scitotenv.2015.10.109, 2016.*

**Some other minor suggestions are reported in the commented MS in attachment.**

Thank you, we addressed all your suggestions, point by point:

**L44: Here, I suggest adding shortly the main results of these studies.** Added a short summary of results of those studies (L44-46)

*"These studies highlight that, in the current Spanish Mediterranean scenario of decrease of total amounts of precipitation but increase in intensity (Serrano-Notivoli et al., 2018), hydrological connectivity is more dependent on rain intensity."*

*Serrano-Notivoli, R., Beguería, S., Saz, M.A., de Luis, M.: Recent trends reveal decreasing intensity of daily precipitation in Spain. Int. J. Climatol., 38(11), 4211-4224, https://doi.org/10.1002/joc.5562, 2018.*

**L49: Could you express a research hypothesis?** Thank you for the suggestion. Now, the research hypothesis is stated in a few lines, closing the introduction section. (L51-53)

*"Based on the watershed physical and climatic characteristics, we hypothesise that rainfall-runoff relationships are based in the intensity and magnitude of singular events, strongly dependent on seasonality of precipitation."*

**L93: What do you mean for "reliable"? Please be more specific.** Thank you. We agree that the term can be confusing and we have removed it without changing the meaning of the sentence.

**L96: How do you ensure this reliability?** We replaced *"reliable"* by *"average"*.

**L113: Not clear why you summed the hourly maximums.** We added an explanation to make clear the reason of summing hourly maximums. (L120)

*"to be representative of the amount of precipitation corresponding to the hours of maximum rainfall"*

**L128: Or "return interval"?** We computed the "return levels" for different "recurrence intervals" (or "return periods"). This paragraph was completely rewritten due to a change in the method of frequency analysis proposed by Referee #1.

*To contextualize the different thresholds of the RE for different probabilities of generating flow in both watersheds, we estimated the return levels of the RE using the generalized Pareto distribution (GPD) for extreme events using the peak-over-threshold (POT) approach. POT is most suitable when complete time series (as RE) are available due to all values exceeding a certain threshold can serve as basis for model fitting (Coles, 2001). We used four different estimators to fit the POT data to a GPD (Maximum Likelihood Estimation (MLE); Unbiased Probability Weighted Moments (PWMU); Moments (MOM); and Likelihood Moment (LME)) to establish proper and wide confidence levels in the estimate of maximum rainfall per RE. Thresholds for the asymptotic approximation by a GPD in both watersheds were manually selected through the graphical representation of Mean Residual Life, the Dispersion index and the scale and shape parameters (see Figure S1 and S2).*

**L159: Significantly?** Modified as suggested.

**Figure 4: Better "Study period".** Modified as suggested.

**L171: Please use a more technical term. Perhaps "quicker" or "higher"?** The expression was changed by the term "faster".

**L201: "the majority"** Modified as suggested.

**L215-220: All this part is methodological and should be moved in that section.** Thank you for your suggestion. Indeed, this part better fits in methodological section, and we moved it accordingly.

**L233-234: I ask the authors to reconsider whether this part may be moved to the methodological section.** Thank you. We moved this part to the methodological section.

**Figure 8: Better to reduce the lables on y-axis.** This figure completely changed to show the results based on a different method of frequency analysis calculation based on a suggestion from Referee #1. Now, labels in Y-axis are better readable.

**L299-302: The location of this part may be also reconsidered.** Thank you for your suggestion. Instead of moving this part of the text, which fits in the linear–non-linear comments of the discussion section, we rewrote it to promote a more fluid reading (L379-380).

*"For this reason, we used a GAM approach, that takes advantage of non-linear relationships, which are highly representative of the great irregularity of precipitation in the Mediterranean area. This approach represents an advantage among the wide variety of methods that has been previously used to model these thresholds in ephemeral or low-yield streams such as multivariate regressions, machine learning approaches, etc. (e.g., Kaplan et al., 2020; Kampf et al., 2018; Shortridge et al., 2016). Furthermore, GAMs allow for avoiding stationarity assumptions in rainfall-runoff relationships (Tian et al., 2020) in comparison with the abovementioned methods."*

**L308: Please reconsider the form of this sentence.** Thank you. Based on the new explanations of the POT method now used for frequency analysis, this part of the text has been slightly changed. Now, the sentence is clearer and more informative (L396-398):

[revised manuscript text omitted]

---

## Author Response (AR2)

**RESPONSE TO REVIEWERS**

**Review of Manuscript No.: hess-2021-352**

**Title: "Rainfall-runoff relationships at event scale in western Mediterranean ephemeral streams"**

**Authors:** Roberto Serrano-Notivoli, Alberto Martínez-Salvador, Rafael García-Lorenzo, David Espín-Sánchez, and Carmelo Conesa-García

We are grateful for the timely comments from the reviewer and the editor to improve the manuscript. In this Response to the Reviewers' letter, we address all the comments and provide extended explanations about the methodological issues raised during the review process. We agree that some procedural aspects were not clearly explained, and we hope that this response letter will help to make clear the statistical analysis.

We used the following color code to facilitate the interpretation of the changes compared to the previous round of revisions:

Questions and responses of the previous round are in light grey text, before the new questions and comments.

**New questions and comments by reviewer and editor are in blue, bold, text [between brackets]. They have been numbered for an easier identification within this response letter.**

New responses are in black normal text. *Addons and modifications in the manuscript are in italic text (only new references are indicated).*
* * *
0. **Editor comment: [How did you avoid including several rainfall amounts of the same event in the analysis (avoid clusters, select independent events, important for the dispersion plot)? How did you use all different diagnostic plots to choose the threshold (reference, explanation)?]**

    Regarding the procedure of obtaining precipitation series to build RE characteristics, response **[19]** provides a detailed explanation and identify the corresponding text changed in the manuscript.

    We added extended explanations, supported by literature references, about the importance of the three studied variables of the RE (duration, magnitude, and intensity) and about the use of the diagnostic plots for threshold selection. The specific responses can be seen in **[4]**, **[10]** and **[11]**.
* * *
Referee #1

This paper aims at better understanding rainfall-runoff relationships through statistical modelling in two ephemeral streams in Spain (with a focus on rainfall events triggering runoff). The paper is well structured. The objectives are also clearly presented.

Evapotranspiration is probably another driver (see L180) – depending when extreme events occur, response in terms of runoff may differ with the stage of plant growth. Why have you not introduced ET0 data (e.g. https://essd.copernicus.org/articles/11/1917/2019/) in your analyses? e.g. considering P- ET0 as explanatory variable.

Thank you for your suggestion. We, indeed, considered the inclusion of ET0 which we are sure that could work as one of the main drivers, especially in summertime as mentioned all along the manuscript. However, the suggested dataset does not fit with our approach of rainfall events isolation. The SPETO dataset is at a weekly temporal resolution that considers the division of a month in 4 periods, always starting in day 1 and aggregating the last days (29/30/31) in the 4th week depending on the month. This approach avoids combined weeks among consecutive months. The weekly aggregation, although useful for climatic analysis, is not applicable to our study, where

we aggregate rainfall events based on daily precipitation data. Additionally, the dataset ends in 2014, meaning that the last 6 years of our period of analysis are not available.

1. **[Why do you not have any other data set? Could you at least say why you cannot use ET0? How do you plant to mention this in the manuscript?]**

Thank you. We did not mention other datasets in the first response because the reviewer asked for that specific publication which, besides, is the only available observational-based product of ETo for Spain. In addition, there are several problems with this variable for our analysis: ETo calculation requires several climatic variables (wind, solar radiation, humidity, etc.) that are not available at the precipitation observatories of the study area in the considered period. Also, we chose not to use theoretical variables such as ETo because we were looking for the particular contribution of the highly irregular and scarce climatic variable of rainfall at event scale on the runoff generation, only using precipitation observations to create a reliable model representing that contribution. We are aware that ETo and many other variables (e.g. soil properties, lithological aspects, etc.) are responsible for a variable part of the runoff (this was partly discussed in the manuscript and we now included new arguments), but that was not the in the focus of the research since we were interested in which part corresponds to rainfall events using observed data.

To make this point clear, we added some explanations to the discussion section:

*"[...] The atmospheric evaporative demand measured in terms of reference evapotranspiration is well known to be a useful climatic factor modelling runoff (Gallart et al., 2002; Goulden and Bales, 2014; Roy et al., 2017). However, we did not use it in our analysis because we pursued unravelling the particular contribution of rainfall, at event scale, on the runoff generation only using precipitation observations to create a reliable model representing that contribution. [...]"*

*Gallart, F., Llorens, P., Latron, J., and Regüés, D.: Hydrological processes and their seasonal controls in a small Mediterranean mountain catchment in the Pyrenees, Hydrol. Earth Syst. Sci., 6, 527–537, https://doi.org/10.5194/hess-6-527-2002, 2002.*

*Goulden, M.L. and Bales, R.C.: Mountain runoff vulnerability to warming, PNAS, 111(39), 14071–14075, https://doi.org/10.1073/pnas.1319316111, 2014.*

*Roy, T., Gupta, H. V., Serrat-Capdevila, A., and Valdes, J. B.: Using satellite-based evapotranspiration estimates to improve the structure of a simple conceptual rainfall–runoff model, Hydrol. Earth Syst. Sci., 21, 879–896, https://doi.org/10.5194/hess-21-879-2017, 2017.*

**I have some doubts about the method used for the frequency analysis: obviously, all the episodes have been kept (more than one value sampled each year) and the peak over threshold approach should be carried out to derive return levels. The generalized Pareto distribution is the most suited distribution (instead of GEV adapted for the block maxima method). For example, the empirical return period of the observed maximum and the length of the time series should be in the same order while Figure 8 suggests return periods > 100 years. Consequently, the rainfall events triggering runoff are probably more frequent than those derived from the frequency analysis. The authors have applied the block maxima approach to data resulting from the selection of over-threshold values (threshold = 0). The method and the discussion should both be revised.**

Thank you for your useful comments. Your argument is right, and we have changed the method to calculate return periods through a *peak-over-threshold* approach. As we now state in methods section (L143-150), this is the most suitable approach due to continuous series of rainfall events are available for both watersheds:

[revised manuscript text omitted]

2. **Figure S1. [we miss any explanation here on what you did and how you changed the manuscript]**

The approach to calculate the return periods was completely changed, we followed the recommendations of the reviewer to avoid the "block maxima" approach and use the "peak-over-threshold" (POT) method. In the previous response, we added a succinct explanation about the used methodology in sections 2.2. and 3.4, new references, and two new figures in the supplementary material (Figures A1 and A2). Figure 8 was also renewed based on the new results and other recommendations by the reviewers. We agree that this part, without being the central focus of the paper, is important to contextualize the return periods and deserves further explanations. For this reason, we re-edited the text of the methodological section to elaborately explain how the POT analysis was performed and how we used the different diagnostic plots to select the thresholds, including references.

*"To contextualize the RE required for different probabilities of generating flow in both watersheds, we estimated the return levels of their magnitude and maximums using a peak-over-threshold (POT) approach. POT is most suitable when complete time series (as RE) are available due to all values exceeding a certain threshold can serve as basis for model fitting (Coles, 2001). The objective was to estimate the return levels of magnitude and maximums of RE for different*

*return periods. The POT method consists in fitting the RE observations higher than a specific threshold to a Generalized Pareto Distribution (GPD). The selection of this threshold must help to subset the appropriate number of observations to reduce the variance without choosing a too low threshold that could induce bias (Ribatet, 2007). In this case, the threshold was derived from the graphical representation of four parameters derived from the RE data: 1) the Mean Residual Life, which shows the mean value of observations over a threshold (mean excess). It is expected to be linear over the threshold at which GPD becomes valid (Acero et al., 2018); 2) the Dispersion Index, which is the ratio between variance and mean of the values over a threshold, with an ideal theoretical value of 1; and 3) the modified scale and 4) shape parameters against a range of thresholds. The parameter estimates (3 and 4) are stable above the threshold at which the GPD model becomes valid. While interpretation of the plots is not always easy, we selected the appropriate thresholds (Figure A1 and A2) based on their convergence to the optimal values of the four graphical representations, as done in similar situations in previous works (Anagnostopoulou and Tolika, 2012; Zakaria et al., 2017).*

*Once thresholds were defined, we used four different estimators to fit the POT data to a GPD (Maximum Likelihood Estimation (MLE); Unbiased Probability Weighted Moments (PWMU); Moments (MOM); and Likelihood Moment (LME)) to establish proper and wide confidence levels in the estimate of maximum rainfall per RE.”*

*Acero, F.J., Gallego, M.C., García, J.A., Usoskin, I.G., Vaquero, J.M.: Extreme Value Theory Applied to the Millennial Sunspot Number Series. Astrophys. J., 830, 80, https://doi.org/10.3847/1538-4357/aaa406. 2018.*

*Anagnostopoulou, C., Tolika, K.: Extreme precipitation in Europe: statistical threshold selection based on climatological criteria. Theor Appl Climatol 107, 479–489, https://doi.org/10.1007/s00704-011-0487-8, 2012.*

*Ribatet, M.: POT: Modelling Peaks Over a Threshold. R News 7(1):34-35. 2007*

*Zakaria, R., Radi, N.F.A., Satari, S.Z.: Extraction method of extreme rainfall data. IOP Conf. Series: Journal of Physics: Conf. Series, 890, 012154, , https://doi.org/10.1088/1742-6596/890/1/012154. 2017.*

3. **[what is on the x-axis in all these graphs?]**

The x-axis represents the range of thresholds in mm. It is now properly labeled in the Figure, thank you.

4. **“Mean Residual Life” [the axis says mean excess]**

The Mean Residual Life represents the mean value of observations over a threshold, which is usually referred as mean excess, as shown in Ribatet (2007). We included this description in the caption of the figure and an extended explanation in the methods section (see previous comment about Figure 1):

*“[…] the threshold was derived from the graphical representation of four parameters derived from the RE data: 1) the Mean Residual Life, which shows the mean value of observations over a threshold (mean excess). It is expected to be linear over the threshold at which GPD becomes valid (Acero et al., 2018) […]”*

5. **“magnitude of 158.3 mm” [of what?]**

The magnitude is referred to the sum of precipitation of the rainfall event (see section 2.2.). We changed this sentence to avoid misunderstandings:

*“[…] required a 158.3 mm magnitude event (sum of total precipitation), […]”*

6. **“the return period is dramatically reduced with probabilities” [revise wording, probabilities are reduced with return period??? you mean "flow generation probability" is reduced with rainfall return period]**

You are right, the sentence is confusing. We changed the text to:

*"[…] the return period is dramatically reduced with low flow generation probabilities […]"*

7. **"The required 3.8 mm of cumulated hourly maximums in Upper Mula to ensure the flow generation at 95% probability are below the selected threshold" [this does not make any sense] "However, the great variability of this model increased the probabilities until 98.8% with a maximum of 44.6 mm, which represents a return period higher than 150 years." [I do not understand what this means] "This large difference reveals the extreme irregularity of flows in Mula and the high uncertainty in prediction based only on the RE." [the difference between what and what?]**

There is maybe a misunderstanding in what we did in our analysis. The rainfall-runoff relationships were analysed in both watersheds based on:

1) The modeling of the rainfall-runoff response to extract what variables (duration, magnitude, or hourly maximums) are important to create new flow, and how much of these variables are required to ensure this new flow at different probabilities.

2) The calculation of the return periods of the extreme values (peaks over threshold) of these important variables (i.e., magnitude for Algeciras and maximums for Upper Mula) to see in which return periods are the highest probabilities of flow generation in each case.

Results of the model showed that with 3.8 mm of hourly maximums in Upper Mula, flow can be generated at 95% probability, however, the threshold selected for the POT analysis to calculate return periods was 7 mm (higher than 95% probability). This means that it is not difficult to ensure flow generation in Upper Mula at 95% probability (actually, almost every year is possible and not very extreme rainfall event is needed). However, if we want to be sure of runoff generation at the highest probability (the model can predict a maximum of 98.8%, meaning a 44.6 mm hourly maximum event), the return period is 150 years. These results showed that the rainfall events required to ensure flow generation with high probabilities (> 95%) are extremely variable (with return periods from 1 to 150 years).

As this part is of key importance for the main message of the manuscript, we rewrote it to make clear how we performed the analysis and what is the meaning of the results. Section *"2.2. Statistical analyses at event scale"* was divided in two subsections (*"2.2.1 Rainfall-runoff modelling"* and *"2.2.2 Return periods of highest probabilities of flow generation"*) to separate the explanations of both parts of the analysis. A synthetic summary was included before them:

*"[…] These variables were used to model the required characteristics of a RE to generate new flow at different probabilities on both watersheds based on:*

> *1)     the modeling of the rainfall-runoff response to identify which variables (duration, magnitude, or hourly maximums) and to what extent they contributed to flow generation at different probabilities; and*

> *2)     the calculation of the return periods of these contributing variables to estimate the likelihood of occurrence of (highest probabilities) of flow generation."*

Results section (*"3.4 Return periods of RE"*) was also modified to include additional explanations about the meaning of the return periods for each watershed.

*"These thresholds mean that all RE in Algeciras with magnitudes lower than 25 mm, and all RE in Mula with cumulated hourly maximums lower than 7 mm, can occur every year and, therefore, the probability of flow generation at 95% in both watersheds has a return period lower than 1 year. However, the RE ensuring the flow generation at a probability higher than 98% span return periods from 4 to >200 years. This large difference in the return periods reveals the extreme irregularity of flows in Mula and the high uncertainty in prediction based only on the RE."*

8. **Figure 8: [Unit of return levels?] / [not clear why the flow probability number should show up in this curve]**

We changed the y-labels to show the units of return levels (mm) and removed the numbers showing the probabilities.

9. **"referred to the fixed character of parameters over time" [assumption of statistics property of the natural processes being constant in time]**
**"climatic series are not stationary" [the output of the climate model is probably stationary, unless you used transient model runs; what you mean is that under climate change, rainfall cannot be considered as being stationary?]**

> We removed this sentence since it was confusing and did not provide a worthwhile contribution to discussion.

There are many studies on rainfall-runoff relationships in ephemeral streams. The authors should develop more the peculiarities of their findings for the two catchments regarding these relationships.

> Thank you for your suggestion. We included several more references to improve the discussion of the results (L359-363; L373-376)
>
> *"[...] For instance, Camarasa (2021) showed that runoff is more dependent on rainfall intensity in the Mediterranean area, and Gutiérrez-Jurado et al. (2019) demonstrated that soil type has the greatest influence on flow generation, as well as Bull et al. (2000) mentioned in a study of a watershed near to our study area. In addition, anthropic interventions such as irrigation, industrial uses, roads, or any water resources change at large scale, can modify rainfall-runoff dynamics, leading to increased consequences of flooding (Conesa-García et al., 2016; Betancourt-Suárez et al., 2021)."*
>
> *"However, a change in the seasonality of flows is expected under these changing conditions of precipitation, leading to potential alterations that could intensify wet and dry periods (Pumo et al., 2016). In Algeciras and Upper Mula watersheds, climate change scenarios also depict a decrease in water resources caused by the changing seasonality, due to an increased evapotranspiration situation (Martínez-Salvador, et al., 2021)."*
>
> *Betancourt-Suárez, V., García-Botella, E., Ramón-Morte, A.: Flood mapping proposal in small watersheds: A case study of the rebollos and miranda ephemeral streams (cartagena, Spain). Water, 13(1), 102, https://doi.org/10.3390/w13010102, 2021.*
>
> *Bull, L.J., Kirkby, M.J., Shannon, J., Hooke, J.M.: The impact of rainstorms on floods in ephemeral channels in southeast Spain. Catena, 38(3), 191-209, https://doi.org/10.1016/S0341-8162(99)00071-5, 2000.*
>
> *Camarasa, A.: Flash-flooding of ephemeral streams in the context of climate change. Geog. Res. Lett., 47(1), 121-142, https://doi.org/10.18172/cig.4838, 2021.*
>
> *Conesa-García, C., García-Lorenzo, R., Pérez-Cutillas, P.: Flood hazards at ford stream crossings on ephemeral channels (south-east coast of Spain). Hydrol. Process., 31(3), 731-749, https://doi.org/10.1002/hyp.11082, 2016.*
>
> *Gutiérrez-Jurado, K.Y., Partington, D., Batelaan, O., Cook, P., Shanafield, M.: What Triggers Streamflow for Intermittent Rivers and Ephemeral Streams in Low-Gradient Catchments in Mediterranean Climates. Water Resour. Res., 55(11), 9926-9946, https://doi.org/10.1029/2019WR02504, 2019.*
>
> *Pumo, D., Caracciolo, D., Viola, F., Noto, L.V.: Climate change effects on the hydrological regime of small non-perennial river basins. Sci. Total Environ., 512(A), 76-92, https://doi.org/10.1016/j.scitotenv.2015.10.109, 2016.*

10. **[we lack reference to statistical analysis in ephemeral streams, what models were used, what explicative variables etc. the current methods have not reference to published work in this field or for the choice of the explicative variables; here you add a mini-discussion on studies in the Mediterranean but no mention of methods used]**

> Thank you for noticing the need of more references and further discussion about the methods addressed in similar works to put in context our research. We added a methodological discussion, properly referenced, to improve the argumentation and the coherence of the manuscript. The new text was included in the discussion section:
>
> *"[...] Most of the previous works based on rainfall-runoff modelling in ephemeral streams were dedicated to runoff forecasting based on rainfall and topographical characteristics at different temporal and spatial scales. Many of these studies used different methods such as transfer-function*

*models (Camarasa et al., 2002), artificial neural networks (Daliakopoulos and Tsanis, 2016; Ahmadi et al., 2019), or hydraulic models (Berardi et al., 2013; Doglioni et al., 2015), amongst others. While they fall into the categories of conceptual or physics-based models (Wheater et al., 1993), our focus is a metric approach using rainfall observations at event scale to characterize the response of flow generation. To this end, we used a GAM method instead of other regression procedures because of its ability to handle nonlinear relationships between the response variable (flow generation) and the set of explanatory variables (Paillex et al., 2019). GAM models have been already used to model rainfall-runoff relationships in ephemeral streams (e.g., van Ogtrop et al., 2011; García-Galiano et al., 2015; Rashid and Beechman, 2019), and they are highly appropriate for these semi-arid environments since they involve the usual highly nonlinear relationships between rainfall and runoff in this type of intermittent rivers (Ye et al., 1997; Goodrich et al., 1997). However, the novelty of our research is vested in the use of the characteristics of rainfall events (duration, magnitude, and maximums) as explanatory variables, instead of the conventional analysis using all rainfall observations (daily, monthly, or annual) without our proposed distinction. Our approach allows to separate the rainfall-runoff responses by the occurrence of rainfall events (consecutive rainy days), avoiding inconsistencies in flow generation of consecutive rainy days due to potential lags between rainfall at headwaters and flow at gauges in lowlands. While the event scale is not new in ephemeral streams studies, most of the event-based analyses are referred to experimental designs based on single or a few events, and/or in sub-daily scales (e.g., De Boer, 1992; Bull et al., 2000; Gutierrez-Jurado et al., 2019). By isolating the rainfall events from daily data over a long period, we provide a general overview of the response of runoff to rainfall. The selection of the explanatory variables was based on the core characteristics of a RE: duration, magnitude (sum of precipitation in the total duration of the event), and intensity (through the sum of hourly maximums). These three variables have been widely used in rainfall-runoff modelling of ephemeral streams (e.g., Camarasa et al., 2002; Kirkby et al., 2005; Hooke, 2016) and represent the rainfall characteristics influencing on runoff generation (Martínez-Mena et al., 1998; Ran et al., 2012; dos Santos, 2017). [...]"*

*Ahmadi, M., Moeini, A., Ahmadi, H., Motamedvaziri, B., Zehtabiyan, G.R.: Comparison of the performance of SWAT, IHACRES and artificial neural networks models in rainfall-runoff simulation (case study: Kan watershed, Iran), Phys. Chem. Earth, 111, 65–77, https://doi.org/10.1016/j.pce.2019.05.002, 2019.*

*Berardi, L., Laucelli, D., Simeone, V. and Giustolisi, O.: Simulating floods in ephemeral streams in Southern Italy by full-2D hydraulic models, Int. J. River Basin Manag., 11(1), 1–17, https://doi.org/10.1080/15715124.2012.746975, 2013.*

*Daliakopoulos, I.N. and Tsanis, I.K.: Comparison of an artificial neural network and a conceptual rainfall–runoff model in the simulation of ephemeral streamflow, Hydrol. Sci. Journal, 61(15), 2763–2774, https://doi.org/10.1080/02626667.2016.1154151, 2016.*

*De Boer, D.H.: Constraints on spatial transference of rainfall-runoff relationships in semiarid basins drained by ephemeral streams, Hydrol. Sci. Journal, 37(5), 491–504, https://doi.org/10.1080/0262666920949261, 1992.*

*dos Santos, J.C.N., de Andrade, E.M., Medeiros, P.H.A., Guerreiro, M.J.S., Araújo, H.: Effect of Rainfall Characteristics on Runoff and Water Erosion for Different Land Uses in a Tropical Semiarid Region, Water Resour. Manage., 31, 173–185, https://doi.org/10.1007/s11269-016-1517-1, 2017.*

*Garcia Galiano, S. G., Olmos Gimenez, P. and Giraldo-Osorio, J. D.: Assessing Nonstationary Spatial Patterns of Extreme Droughts from Long-Term High-Resolution Observational Dataset on a Semiarid Basin (Spain), Water, 7(10), 5458–5473, https://doi.org/10.3390/w7105458, 2015.*

*Martínez-Mena, M., Albaladejo, J., and Castillo, V. M.: Factors influencing surface runoff generation in a Mediterranean semi-arid environment: Chicamo watershed, SE Spain, Hydrol. process., 12(5), 741–754, https://doi.org/10.1002/(SICI)1099-1085(19980430)12:5%3C741::AID-HYP622%3E3.0.CO;2-F, 1998.*

*Paillex, A., Siebers, A.R., Ebi, C., Mesman, J., Robinson, C.T.: High stream intermittency in an alpine fluvial network: Val Roseg, Switzerland, Limnol. Oceanogr., 65(3), 557–568, https://doi.org/10.1002/lno.11324, 2019.*

*Ran, Q., Su, D., Li, P., He, Z.: Experimental study of the impact of rainfall characteristics on runoff generation and soil erosion, J. Hydrol., 424–425, 99–111, https://doi.org/10.1016/j.jhydrol.2011.12.035, 2012.*

*Rashid, M. and Beecham, S.: Simulation of streamflow with statistically downscaled daily rainfall using a hybrid of wavelet and GAMLSS models, Hydrol. Sci. Journal, 64:11, 1327–1339, https://doi.org/10.1080/02626667.2019.1630742, 2019.*

*Wheater H.S., Jakeman A.J. and Beven K.J.: Progress and directions in rainfall–runoff modelling. In Jakeman A.J., Beck M.B., and McAleer M.J., editors, Modelling Change in Environmental Systems, pages 101–132. John Wiley & Sons, Chichester, UK, 1993.*

**11. [this paragraph does not mention the word ephemeral streams; we do not get any additional insights on how relationships between rainfall and runoff might look like for ephemeral streams] "as well as Bull et al. (2000) mentioned in a study of a watershed near to our study area" [incomplete sentence]**

The complete paragraph in the discussion section, including the incomplete sentence, has been re-written to make clear the content, and properly reference the ephemeral streams.

*"For instance, Camarasa (2021) showed that runoff in ephemeral streams is more dependent on rainfall intensity in the Mediterranean area than in non-arid environments, and Gutiérrez-Jurado et al. (2019) and Bull et al. (2000) showed that soil type has the greatest influence on flow generation in intermittent rivers. In summary, rainfall-runoff relationships in ephemeral streams are influenced by topography and soil characteristics (Wooldridge et al., 2003; Chen et al., 2019), however, their flows are heavily dependent on the intensity, which is usually considered as the ratio between the volume of rainfall (magnitude) in a RE and its duration (e.g., Camarasa and Tilford, 2002; La Torre Torres et al., 2011; El Afy, 2016). In addition to the topographical and climatic characteristics of the watersheds, anthropic interventions such as irrigation, industrial uses, roads, or any water resources change at large scale, can modify rainfall-runoff dynamics, leading to increased consequences of flooding (Conesa-García et al., 2016; Betancourt-Suárez et al., 2021)."*

*Chen, S. A., Michaelides, K., Grieve, S. W., & Singer, M. B.: Aridity is expressed in river topography globally. Nature, 573(7775), 573–577, https://doi.org/10.1038/s41586-019-1558-8, 2019.*

*Wooldridge, S. A., Kalma, J. D., and Walker, J. P.: Importance of soil moisture measurements for inferring parameters in hydrologic models of low-yielding ephemeral catchments, Env. Mod. Soft., 18(1), 35–48, https://doi.org/10.1016/S1364-8152(02)00038-5, 2003.*

*La Torre Torres, I.B., Amatya, D.M., Sun, G., and Callahan, T.J.: Seasonal rainfall–runoff relationships in a lowland forested watershed in the southeastern USA, Hydrol. Process., 25(13), 2032–2045, https://doi.org/10.1002/hyp.7955, 2011.*

*El Alfy, M.: Assessing the impact of arid area urbanization on flash floods using GIS, remote sensing, and HEC-HMS rainfall–runoff modelling, Hydrology Research, 47(6), 1142–1160, https://doi.org/10.2166/nh.2016.133, 2016*

Details:

L35: There is an inversion between first name and last name in the reference « Thibault et al. 2017 ». = = > Datry et al. is the correct reference.

Modified as suggested.

L40: a reference regarding sediment transport: https://doi.org/10.1016/j.catena.2020.104865

Reference added.

Fig. 1: we do not see the main river network. Please add the location of the two reservoirs, even if we guess that they are the mouths of the two catchments and point out the stations used to compute the precipitation time series.

Modified as suggested.

**L102-106: The authors used long time series to perform a stationarity analysis. Are gridded and local data consistent during the concomitant period (correlation, mean, etc.)? This is important to assess the representativeness of the gridded data for the two catchments.**

The SPREAD dataset, referenced work as Serrano-Notivoli et al. (2017), spans the period from 1950 to 2012. It was extended until 2020 in the study area using the same data series as used in the rest of the analysis, through the method described in Serrano-Notivoli et al. (2017b) to ensure the reliability of the data. We have added this reference to make clear this point in the methodological section (L110-112).

*"[…] we used the SPREAD dataset (Serrano-Notivoli et al., 2017), a daily gridded precipitation dataset covering the whole Spanish territory at a 5x5 km spatial resolution, to analyse long-term trends of annual precipitation of the two watersheds by extending its period coverage until 2020 following Serrano-Notivoli et al. (2017b)."*

*Serrano-Notivoli, R., de Luis, M. and Beguería, S.: An R package for daily precipitation climate series reconstruction. Environ. Modell. Softw., 89, 190-195, http://dx.doi.org/10.1016/j.envsoft.2016.11.005, 2017b.*

**L218-219 & S2: Some criteria have been computed, but not commented (please add some comments or delete the values).**

Thank you for your comments. We moved the table to supplementary material and referenced in the text the table with the GAM summaries for both watersheds.

**Figs 6, 7 and 8: Please use semi-log plots with the y-axis on a logarithmic scale to make the reading easier.**

Thank you for your suggestion. Figure 6 has been changed to show all variables in logarithmic scale. As this action increased some Pearson values, corresponding texts in the manuscript have been adapted to the new results. Figure 7 and (new figure) 8 are already in a semi-log scale.

[Figure]

*Figure 6: Values of precipitation variables and flow contribution (ΔQ) of all events in Algeciras (bottom left side) and Mula (top right side). Magnitude and maximum variables are in logarithmic scale. Pearson correlations are shown in red (all correlations are significant at α<0.01)*

12. **"As this action increased some Pearson values" [how can a change of plotting change the values? say in which space you plotted correlations, they should be in the original space]**

This is a misunderstanding from our response in the previous round: we transformed all the variables to a logarithmic scale to normalize their frequency distributions and to remove their

dependency from the different measuring units (days, mm, m3/s). This action slightly changed some of the Pearson correlation results. Now, the figure shows the same scale for all variables.
* * *
**Referee #2**

**The authors propose a study that analyze the transformation rainfall-runoff in semi-arid catchments of Southern Spain, where the ephemeral regime of rivers and the climatic stress may lead to hazardous floods or, on the contrary, to dramatic droughts. The study is quite novel and gives significant insight about precipitation depths and return intervals, which may determine water and sediment flows in the channels. The statistical analysis is fine and suitable for the study aims. The results are presented with clearness and synthesis. Although the study is of good quality, I have three suggestions that may improve the MS:**

**- several methodological sentences are reported in the results sections, and this may confuse the reader. I ask the authors to revise both parts.**

> Thank you for your suggestion. We moved the methodological descriptions in results section to methodology (see detailed lines and paragraphs at the end of this document).

**- although literature about the flow regime in ephemeral channels is not abundant, some other cross-comparisons with similar studies may further valorize the study results**

> Thank you. We added several new references to improve the discussion in all the addressed aspects and to compare our work with similar research in nearby areas and in similar terms (L359-363).

> *"[...] For instance, Camarasa (2021) showed that runoff is more dependent on rainfall intensity in the Mediterranean area, and Gutiérrez-Jurado et al. (2019) demonstrated that soil type has the greatest influence on flow generation, as well as Bull et al. (2000) mentioned in a study of a watershed near to our study area. In addition, anthropic interventions such as irrigation, industrial uses, roads, or any water resources change at large scale, can modify rainfall-runoff dynamics, leading to increased consequences of flooding (Conesa-García et al., 2016; Betancourt-Suárez et al., 2021)."*

> *Betancourt-Suárez, V., García-Botella, E., Ramón-Morte, A.: Flood mapping proposal in small watersheds: A case study of the rebollos and miranda ephemeral streams (cartagena, Spain). Water, 13(1), 102, https://doi.org/10.3390/w13010102, 2021.*

> *Bull, L.J., Kirkby, M.J., Shannon, J., Hooke, J.M.: The impact of rainstorms on floods in ephemeral channels in southeast Spain. Catena, 38(3), 191-209, https://doi.org/10.1016/S0341-8162(99)00071-5, 2000.*

> *Camarasa, A.: Flash-flooding of ephemeral streams in the context of climate change. Geog. Res. Lett., 47(1), 121-142, https://doi.org/10.18172/cig.4838, 2021.*

> *Conesa-García, C., García-Lorenzo, R., Pérez-Cutillas, P.: Flood hazards at ford stream crossings on ephemeral channels (south-east coast of Spain). Hydrol. Process., 31(3), 731-749, https://doi.org/10.1002/hyp.11082, 2016.*

> *Gutiérrez-Jurado, K.Y., Partington, D., Batelaan, O., Cook, P., Shanafield, M.: What Triggers Streamflow for Intermittent Rivers and Ephemeral Streams in Low-Gradient Catchments in Mediterranean Climates. Water Resour. Res., 55(11), 9926-9946, https://doi.org/10.1029/2019WR02504, 2019.*

**13. **[no reference to ephemeral streams]**

> See response **[11]**. This paragraph has been completely re-written to include new references, to focus the argumentation into the ephemeral streams' discussion, and to complete the contextualization of the rainfall-runoff relationships within the recent contributions of literature.

**- some expectations about the future trends of rainfall-runoff transformation under the forecasted climate change scenarios (higher mean temperature and lower precipitation amounts) in the studied area may be reported on the authors' knowledge and experience.**

> Regarding future trends, we added a few lines relating the rainfall-runoff potential changes in both watersheds to the expected decrease in precipitation and increase in temperature, leading to a higher evapotranspiration. As stated in previous works, this scenario, depending on the emissions development, has high probabilities of produce changes in seasonality of flows, increasing risks

of floods and droughts. Despite we do not explicitly address climate change scenarios in our study, we appreciate your comment and agree that it deserves to be mentioned (L373-376).

*"However, a change in the seasonality of flows is expected under these changing conditions of precipitation, leading to potential alterations that could intensify wet and dry periods (Pumo et al., 2016). In Algeciras and Upper Mula watersheds, climate change scenarios also depict a decrease in water resources caused by the changing seasonality, due to an increased evapotranspiration situation (Martínez-Salvador, et al., 2021)."*

*Martínez-Salvador, A., Millares, A., Eekhout, J.P.C. and Conesa-García, C.: Assessment of Streamflow from EURO-CORDEX Regional Climate Simulations in Semi-Arid Catchments Using the SWAT Model. Sustainability, 13(13), 7120, https://doi.org/10.3390/su13137120, 2021.*

*Pumo, D., Caracciolo, D., Viola, F., Noto, L.V.: Climate change effects on the hydrological regime of small non-perennial river basins. Sci. Total Environ., 512(A), 76-92, https://doi.org/10.1016/j.scitotenv.2015.10.109, 2016.*

**Some other minor suggestions are reported in the commented MS in attachment.**

Thank you, we addressed all your suggestions, point by point:

**L44: Here, I suggest adding shortly the main results of these studies.** Added a short summary of results of those studies (L44-46)

*"These studies highlight that, in the current Spanish Mediterranean scenario of decrease of total amounts of precipitation but increase in intensity (Serrano-Notivoli et al., 2018), hydrological connectivity is more dependent on rain intensity."*

*Serrano-Notivoli, R., Beguería, S., Saz, M.A., de Luis, M.: Recent trends reveal decreasing intensity of daily precipitation in Spain. Int. J. Climatol., 38(11), 4211-4224, https://doi.org/10.1002/joc.5562, 2018.*

**L49: Could you express a research hypothesis?** Thank you for the suggestion. Now, the research hypothesis is stated in a few lines, closing the introduction section. (L51-53)

*"Based on the watershed physical and climatic characteristics, we hypothesise that rainfall-runoff relationships are based in the intensity and magnitude of singular events, strongly dependent on seasonality of precipitation."*

**L93: What do you mean for "reliable"? Please be more specific.** Thank you. We agree that the term can be confusing and we have removed it without changing the meaning of the sentence.

**L96: How do you ensure this reliability?** We replaced *"reliable"* by *"average"*.

**L113: Not clear why you summed the hourly maximums.** We added an explanation to make clear the reason of summing hourly maximums. (L120)

*"to be representative of the amount of precipitation corresponding to the hours of maximum rainfall"*

**L128: Or "return interval"?** We computed the "return levels" for different "recurrence intervals" (or "return periods"). This paragraph was completely rewritten due to a change in the method of frequency analysis proposed by Referee #1.

*To contextualize the different thresholds of the RE for different probabilities of generating flow in both watersheds, we estimated the return levels of the RE using the generalized Pareto distribution (GPD) for extreme events using the peak-over-threshold (POT) approach. POT is most suitable when complete time series (as RE) are available due to all values exceeding a certain threshold can serve as basis for model fitting (Coles, 2001). We used four different estimators to fit the POT data to a GPD (Maximum Likelihood Estimation (MLE); Unbiased Probability Weighted Moments (PWMU); Moments (MOM); and Likelihood Moment (LME)) to establish proper and wide confidence levels in the estimate of maximum rainfall per RE. Thresholds for the asymptotic approximation by a GPD in both watersheds were manually selected through the graphical representation of Mean Residual Life, the Dispersion index and the scale and shape parameters (see Figure S1 and S2).*

**L159: Significantly?** Modified as suggested.

**Figure 4: Better "Study period".** Modified as suggested.

**L171: Please use a more technical term. Perhaps "quicker" or "higher"?** The expression was changed by the term "faster".

**L201: "the majority"** Modified as suggested.

**L215-220: All this part is methodological and should be moved in that section.** Thank you for your suggestion. Indeed, this part better fits in methodological section, and we moved it accordingly.

**L233-234: I ask the authors to reconsider whether this part may be moved to the methodological section.** Thank you. We moved this part to the methodological section.

**Figure 8: Better to reduce the lables on y-axis.** This figure completely changed to show the results based on a different method of frequency analysis calculation based on a suggestion from Referee #1. Now, labels in Y-axis are better readable.

**L299-302: The location of this part may be also reconsidered.** Thank you for your suggestion. Instead of moving this part of the text, which fits in the linear–non-linear comments of the discussion section, we rewrote it to promote a more fluid reading (L379-380).

*"For this reason, we used a GAM approach, that takes advantage of non-linear relationships, which are highly representative of the great irregularity of precipitation in the Mediterranean area. This approach represents an advantage among the wide variety of methods that has been previously used to model these thresholds in ephemeral or low-yield streams such as multivariate regressions, machine learning approaches, etc. (e.g., Kaplan et al., 2020; Kampf et al., 2018; Shortridge et al., 2016). Furthermore, GAMs allow for avoiding stationarity assumptions in rainfall-runoff relationships (Tian et al., 2020) in comparison with the abovementioned methods."*

**L308: Please reconsider the form of this sentence.** Thank you. Based on the new explanations of the POT method now used for frequency analysis, this part of the text has been slightly changed. Now, the sentence is clearer and more informative (L396-398):

*"Low return periods were shown for events generating new flow at 95% probability, but they dramatically increased when probabilities were increased until maximum (99.5% in Algeciras and 98.8% in Mula). However, the analysis has some limitations to consider."*

**14. P2.L43: processes?**

Modified as suggested.

**15. P2.L46: more than on what?**

We changed this sentence to make clear its significance.

*"[…] These studies highlight that, in the current Spanish Mediterranean scenario of decrease of total amounts of precipitation but increase in intensity (Serrano-Notivoli et al., 2018), hydrological connectivity is more dependent on rain intensity than in the past"*

**16. P2.L52-53: what means "relationships are based in"? second: do you mean rainfall intensity and amount? does your hypothesis imply that rainfall could not have an effect? or that only intensity could have an effect or only amount? the hypothesis is not very clear**

Thank you for your comments. We changed this text to explain that we hypothesize that runoff is highly dependent on rainfall intensity and magnitude.

*"[…] Based on the watershed physical and climatic characteristics, we hypothesise that runoff highly depends on the intensity and amount of rainfall of singular events."*

**17. P2.L60-61: negative water balance the entire year? so, when does it fill up, from where? a water balance cannot be negative for long time periods (except if there was a huge groundwater storage that is emptying)**

We agree that this sentence can be confusing. We meant that, overall, the whole region shows theoretical values of negative water balance. We changed the sentence avoid confusions.

*"the evapotranspiration rate is among the highest in Spain (Tomás-Burguera et al., 2020), leading to a negative water balance in the whole region, especially in summer months (June, July, and August) and being highly variable depending on the season and the year."*

**18. P2.L61-62: if high rate of infiltration, we should have high subsurface flow and thus runoff from subsurface flow; what means hampering "runoff" here?**

Thank you for noticing this issue. We meant *"surface runoff"*. It is now changed in the text.

**19. P5.L05-06: the procedure of obtaining precip is not entirely clear; should publish the series along with paper**

As suggested, we included now as supplementary material the RE data series for both watersheds, containing the initial and ending date of the events, their duration, magnitude (total volume of precipitation), maximums (cumulated hourly maximums), and the contributed flow (difference between the cumulated flow during the RE and flow of the day before the RE) (see section 2.2. for details).

Section "*2.1 Data*" details how the regional precipitation series were built for each of the watersheds from the precipitation data of nearest meteorological stations. We averaged the daily rainfall values of the corresponding stations for each watershed to obtain a representative data series of precipitation to relate with flow data series. The rainfall events (RE) series were constructed from these unique daily precipitation series by grouping the consecutive days with rainfall, as described in section "*2.2 Statistical analyses at event scale*". In this regard, each one of the RE includes the volume of precipitation, averaged from the nearest stations. While the

amounts of rainfall of the RE could be different throughout the watershed, this is a representative, average, depiction of the events.

In order to clarify the process, we reformulated the explanation in section 2.1.:

*"[…] The regional series for each watershed were built with 2 variables: 1) the daily average of total precipitation in 24 hours and 2) the daily average of maximum precipitation in 1 hour. With the aim of relating these series with the temporal availability of flow data, they were built for 2003-2020 in Algeciras and for 1996-2020 in Mula. […]"*

**20. P5.L15: we have here no reference to previous work on intermittent flow, as required by the reviewers**

We added an extended discussion about methodological procedures of rainfall-runoff relationships in ephemeral streams in the discussion section. See response **[10]**.

**21. P5.L18: how long has to be the dry period to identify two separate events?**

At least one day with zero precipitation is enough to separate two events. This has been included in the text:

*"Rainfall events (RE) were detected from daily data series for the whole period in both watersheds by grouping consecutive wet days separated, at least, by one dry day (P = 0)."*

**22. P6.L38: (Residual Deviance) what is this? (UBRE) reference?**

We changed this sentence to add extended explanations about these two estimate errors:

*"[…] we compared different models using from one to all variables through two conventional estimate errors (see Table A1): AIC (Akaike Information Criterion) and logLik (log-likelihood), and two specific estimate errors for GAMs: deviance (Residual deviance) and UBRE (Un-Biased Risk Estimator). Residual deviance is defined as twice the difference between the log likelihood of a model that provides a perfect fit (also called the saturated model) for the model under study (Zuur et al., 2009), and the UBRE is essentially a rescaled AIC used to estimate the mean square error on GAMs (Wood, 2017). […]"*

*Wood, S.N.: Generalized Additive Models: An Introduction with R (2nd ed.). Chapman and Hall/CRC. https://doi.org/10.1201/9781315370279, 2017.*

*Zuur A.F., Ieno E.N., Walker N.J., Saveliev A.A. and Smith G.M.: GLM and GAM for Count Data. In: Mixed effects models and extensions in ecology with R. Statistics for Biology and Health. Springer, New York, NY. https://doi.org/10.1007/978-0-387-87458-6_9, 2009.*

**23. P6.L50: reference? and hopefully some more explanation in the appendix**

This part of the text has been changed to extend these explanations. See response **[2]** for details.

**24. P12.L62-63: the result of the model which is the subject of the paper cannot be in an appendix**

Table A3 (now, Table 3) with results of the model has been moved to the main text from the appendix.

**25. P13.L93: evaporation????**

Thank you, we included evaporation as suggested possible driver of flow generation.

*"[…] topographical and soil characteristics, as well as other climatic factors such as evaporation, probably play an important role that is difficult to integrate in these types of models. […]"*

26. **P14.Figure7: say that Proba(Qbin)>0.5 means flow?**

We did not suggest neither in the text nor in the figure that a probability higher than 0.5 means flow. Diagnostic plots (Figure 7) show the partial effects of the GAM model, meaning that they represent the probability of flow generation dependent of each partial effect while the other remains in its average values. It is explained after Table 4.

27. **P14.L17-18: how can you interpret the magnitude only if the model uses duration and magnitude? second: why the 99.5% probability threshold, what does it correspond to?**
**P14.L20: according to the fitted GDP models would correspond to...**
**P14.L20: 50 to 100 not with "-"**

As the RE average duration is similar in both watersheds (1.9 and 2.1 days), as well as their response to this variable in the GAM model (see section 3.3, before Figure 7), we opted to calculate the return periods of the other variables: magnitude for Algeciras and maximums for Upper Mula. The referenced sentence means that the maximum probability of flow generation from the GAM model is able to predict, being the duration in its average value (1.9 days), is 99.5%, meaning a magnitude (total amount of precipitation) of 158.3 mm. This magnitude represents a return period from 50 to 100 years.

*"[...] The maximum probability of flow generation that the GAM model was able to predict for Algeciras, being the duration in its average value (1.9 days), is 99.5%, which corresponds with a RE of magnitude of 158.3 mm (sum of total precipitation). According to the fitted POT values to a GPD, the return period of this magnitude ranged from 50 to 100 years. [...]"*

28. **P14.L20: revise wording, probabilities are reduced with return period??? you mean "flow generation probability" is reduced with rainfall return period**

Modified as suggested. See response [6] since this is the same comment.

29. **P15.L29-30: what does this mean for the statistical analysis, is this a problem or not?**
**P15.L30-31: unclear what you say here**
**P15.L31-32: high uncertainty of the extreme value distribution or of the GAM model?**

We realized that this sentence is not required now for the understanding of the results of this section and we removed it. The 95% probability of flow generation is below the threshold selected by POT in both watersheds and, therefore, this sentence does not provide new insights. Furthermore, this is now clearly explained in the first paragraph of this section (an extended explanation in this regard, and the new text included in the manuscript, can be seen in response [7]). We changed the text in which results for Mula are presented:

*"[...] Similar results were obtained for Mula, where the maximum probability (98.8%) of flow generation implied an RE with a cumulated hourly maximum of 44.6 mm, which represents a return period higher than 150 years. [...]"*

30. **P16.L49-50: as far as I see, we do not see this in the paper since the model is not presented in the main text**

We now included the table with the model outputs in the main text. However, this sentence is better understood when referred to Figure 7 (now referenced in the text), which shows the similar behavior of duration and the different patterns of magnitude and intensity between both watersheds.

31. **P16.L56: what do you mean? how it goes into the subsoil?**
**P16.L58-59: are you talking about climate change here?**

No, we meant that the variability in the RE characteristics drives flow generation, and we realize now that the complete sentence is a bit repetitive. We have changed this part to avoid confusions:

*"[…] Thus, in addition to the dependence on the lithological and terrain configuration (van Dijk, 2010) and changes in seasonal precipitation regimes (Fakir et al., 2021), the RE duration, intensity, and magnitude, have a high probability of changing the available flow, as shown in the results of the GAM model. […]"*

**32. P16.L82: well, here you make the stationarity assumption?**

Not sure what you mean. We are not making assumptions of stationarity, actually we try here to explain the contrary. Linear models assume that the relationship between variables is stationary. On the contrary, GAM models are non-linear and assume, in our case, that relationships of rainfall and runoff are not stationary.

**33. P17.L94: thresholds of what?**

To avoid confusions, we changed *"thresholds"* by *"rainfall characteristics"*.

**34. P17.L05-06: reword, see comments in response file**

Modified as suggested. This sentence has been removed as explained in response **[9]**.

**35. P17.L10-11: not clear from the paper what we could now do with the model**

Thank you but your comment is unclear. If we understand correctly, you are asking for the contributions of the GAM model in this research. This is summarized in the first line of this paragraph:

*"[…] the nonlinear analysis of RE helped to understand the type of events required to generate new flow in both watersheds […]"*

And widely developed in the conclusions section, where the main results are recapitulated:

*"[…] While a linear relationship was insufficient to derive robust conclusions about flow production and rainfall, a nonlinear analysis using GAMs helped to understand that most of the new flow is driven by a similar duration of the rainfall events (4-5 days to ensure a 95% probability) in both watersheds. However, the magnitude of the event (cumulated precipitation) was a more significant predictor in Algeciras (20.7 mm) than Upper Mula, where maximums (cumulated hourly maximums of each day) showed a higher significance (3.8 mm) […]"*

The model helps to understand rainfall-runoff relationships in both watersheds and disentangle the type of rainfall events required for new flow generation in these ephemeral streams. This is also mentioned in the abstract:

*"[…] We explored the rainfall-runoff relationships in two semiarid watersheds in southern Spain (Algeciras and Upper Mula) to model the different types of rainfall events required to generate new flow in both intermittent streams […]"*

**36. P17.L17-18: ok, useful conclusion**

Ok. Thank you for your comment.

**37. P18.L33: significance in mm? you mean magnitude?**

Sorry, the meaning of the sentence was intended to be different. We modified the text:

*"[…] where cumulated hourly maximums of each day (3.8 mm) showed a higher significance than in Algeciras […]"*

**38. P18.L37-39: based on model or on analysis of data?**

We meant the observed data. Additionally, we corrected some typos to be consistent with verbal tenses. We modified the text:

*"[…] Results showed that the precipitation regime was very irregular, and the observed average event of 1.2 days and less than 1.5 mm was clearly insufficient to generate new flow. Almost a third part of the rainfall events were non-contributing for flow generation (flows were similar or lower than previous day to the rainfall event). […]"*

**39. P18.L40-41: reformulate: use "rainfall return periods " and "flow generation probability"**

Modified as suggested.

---

## Author Response (AR3)

**RESPONSE TO REVIEWERS**

**Review of Manuscript No.: hess-2021-352**

**Title: "Rainfall-runoff relationships at event scale in western Mediterranean ephemeral streams"**

**Authors:** Roberto Serrano-Notivoli, Alberto Martínez-Salvador, Rafael García-Lorenzo, David Espín-Sánchez, and Carmelo Conesa-García

Dear editor,

Thank you for your revision. Your argument is right since there was a mistake in the calculation of the return periods. We wrongly assumed one rainfall event per year, but the actual frequency is higher for both watersheds. We recalculated the return periods with the correct frequency and now Figure 8 shows a realistic picture of rainfall events occurrence. We also changed the corresponding values of return periods in section 3.4.